# A Benchmark for Automatic ML Research Agents Highlighting the Importance of Exploration Breadth

## Abstract

Large language models (LLMs) have sparked growing interest in automatic machine learning research agents. Among them, agents capable of autonomously proposing ideas and conducting machine learning experiments are particularly promising, as they maximize research automation and accelerate scientific progress by iteratively refining ideas based on experimental results. However, comprehensively evaluating such agents remains challenging. Existing benchmarks tend to overemphasize engineering aspects while neglecting academic rigor, creating barriers that obscure a clear assessment of an agent's scientific capabilities of machine learning research. They also suffer from limited task diversity, an overemphasis on application-oriented tasks over fundamental research problems, and limited scalability to realistic research settings. To address these limitations, we introduce FML-bench, a benchmark designed to evaluate automatic machine learning research agents on 8 diverse and fundamental machine learning research problems. It reduces coding burden, emphasizes fundamental problems rather than specific use cases, offers high task diversity, and is extensible to real-world machine learning GitHub repository. Furthermore, we present a unified evaluation framework with five complementary metrics, designed to comprehensively assess agent performance on our benchmark. We evaluate state-of-the-art automatic research agents on FML-bench, and find that agents employing broad research exploration strategies outperform those focusing on narrow but deep exploration. These findings suggest that emphasizing the breadth of exploration may lead to more effective research outcomes than focusing solely on incremental refinement. Our benchmark is available at anonymous github: `https://anonymous.4open.science/r/Anonymous-78B6`.

## 1 Introduction

Large language models (LLMs) have catalyzed a resurgence of interest in automatic machine learning (ML) research agents which assist or carry out parts of the scientific discovery workflow. These agents not only support hypothesis generation, coding, and experiment management, but also increasingly act as collaborators in discovery by providing complementary perspectives that can accelerate machine learning research across domains. Within this landscape, agents that automatically propose ideas and run experiments are particularly compelling (Lu et al., 2024; Yamada et al., 2025). They close the loop from ideation to empirical validation to maximize automation and to speed up research cycles. Compared to settings that only elicit ideas and then use LLMs or human to assess "novelty" and "feasibility" which often diverge from real-world utility, this approach evaluates agents based on actual experimental outcomes, providing objective and quantitative evidence of their effectiveness (Wang et al., 2024a; Baek et al., 2024; Si et al., 2024).

Despite rapid progress, existing benchmarks offer an incomplete picture of research competence, as shown in Tab. 1. **First**, most focus on Kaggle-style, usecase-oriented tasks and emphasize engineering execution (e.g., feature engineering, standardized model training, and optimization) while paying limited attention to evaluating an agent's ability to tackle fundamental machine learning research problems, such as representation learning and generalization (Chan et al., 2024; Huang et al., 2023; Padigela et al., 2025; Jing et al., 2024). **Second**, these benchmarks primarily focus their

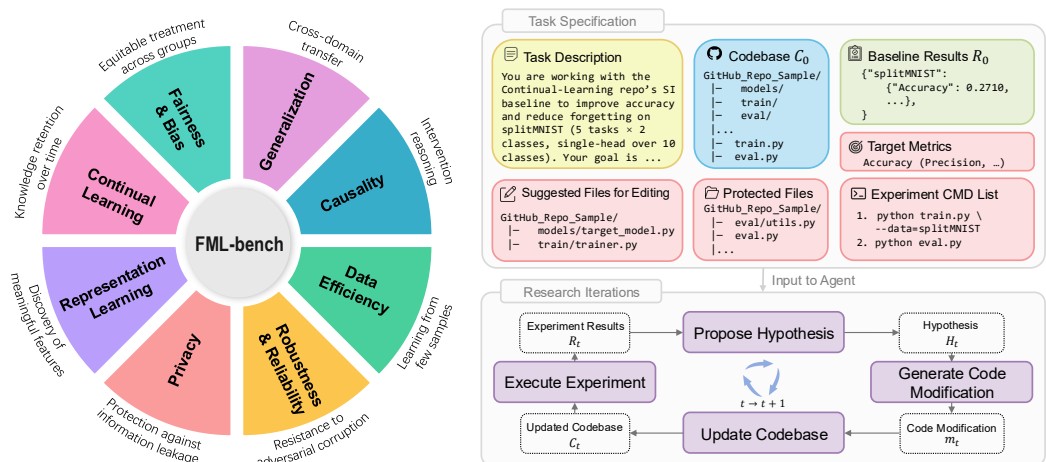

Figure 1: **Overview of FML-bench.** FML-bench includes 8 fundamental machine learning research tasks, designed to evaluate agents' capabilities in solving machine learning research problem. Agents are assessed on their ability to solve machine learning problem through iterative research.

evaluation on final task performance metrics (e.g., accuracy, recall) and computational cost, while overlooking the characteristics of agents' internal iterative processes.

Furthermore, some benchmarks provide only raw data without baseline code (Chan et al., 2024; Jing et al., 2024), making it difficult to systematically assess agents' research capabilities while introducing coding barriers that can obscure academic merit (e.g., when sound ideas fail due to engineering pitfalls). In addition, even when baseline codebases are provided, they are often handcrafted and tightly formatted (Huang et al., 2023; Padigela et al., 2025) which hinders their scalability. This is because adapting them to new tasks usually requires substantial re-engineering to conform to their benchmark design, rather than allowing direct use of existing codebases.

To address these gaps, we introduce FML-bench, a benchmark designed to evaluate automatic ML research agents on fundamental ML problems. FML-bench comprises 8 diverse tasks (Fig. 1) chosen to reflect bottlenecks that repeatedly surface in modern ML. The design follows four principles. **1) Fundamental ML problems.** Designed tasks target core scientific challenges rather than application products or leaderboard scoring, keeping the focus on research questions. **2) Real-world codebases.** Tasks are instantiated from existing research repositories, mirroring typical practice where new ideas are tested by adapting prior code. **3) Extensibility by construction.** The benchmark can easily incorporate machine learning GitHub repositories that support end-to-end training and evaluation, requiring only minor output-format adapters. **4) Low coding barrier.** Agents are not required to build entire codebases from scratch, but can start from provided baselines. This setup enables agents to focus on scientific advances in algorithms and architectures rather than on purely engineering effort.

The tasks included in FML-bench span a broad set of foundational problems (Wang et al., 2022; Pearl, 2019; Adadi, 2021; Goodfellow et al., 2014; Abadi et al., 2016; Bengio et al., 2013; Chen & Liu, 2018; Mehrabi et al., 2021): generalization (cross-domain transfer), data efficiency (learning from few samples), representation learning (discovery of meaningful features), continual learning (knowledge retention over time), causality (intervention reasoning), robustness and reliability (resistance to adversarial corruption), privacy (protection against information leakage) and fairness and bias (equitable treatment across groups). Agents are expected to propose new or improved ML methods that deliver stronger empirical results than baselines across these eight tasks.

To evaluate agents holistically, we provide a systematic evaluation platform enabling researchers to identify and analyze factors influencing agent performance. We formalize five complementary metrics that capture different facets of research competence. Utility measures empirical performance improvement and serves as the primary objective. Diversity quantifies the variety of code modifications proposed. Academic contribution rate distinguishes academic modifications (e.g., new losses, architectures, or training schemes) from engineering modifications (e.g., hyperparameter tuning),

Table 1: Comparison of ML agent benchmarks across key design goals and agent requirements. Repo refers to the repository, and Comp denotes Competition. *: In MLAgentBench, only part of the tasks meet this requirement; users must prepare baseline and evaluation code even when some tasks are based on real-world Kaggle repositories.

| Criterion | Ours | MLE–Bench | MLAgentBench | ML–Dev–Bench | DSBench |
|---|---|---|---|---|---|
| *Design Goals* | | | | | |
| Fundamental ML Problem Focus | ✓ | ✗ | ✗ | ✗ | ✗ |
| Agent Behavior Analysis | ✓ | ✗ | ✗ | ✗ | ✗ |
| Real-World Repo/Comp | ✓ | ✓ | ✓* | ✗ | ✓ |
| Low Coding Barrier | ✓ | ✗ | ✓ | ✓ | ✗ |
| Scalability via Existing Repo/Comp | ✓ | ✓ | ✗ | ✗ | ✓ |
| *Requirements to Agent* | | | | | |
| Understand Codebase | ✓ | ✗ | ✓ | ✓ | ✗ |
| Understand Data | ✗ | ✓ | ✗ | ✗ | ✓ |
| Execute Arbitrary CMD with Args | ✓ | ✗ | ✓ | ✓ | ✗ |
| Execute Multi-step CMD List | ✓ | ✗ | ✗ | ✗ | ✗ |

rewarding agents that prioritize scientifically meaningful changes. Cost accounts for computational and time expenditure. Step success rate captures the fraction of runs that produce valid results without bugs, reflecting an agent's reliability in multi-step workflows. Notably, Diversity and Academic Contribution Rate are specifically proposed to analyze agent behavior throughout the entire research iteration process, providing insights into how agents explore and develop solutions rather than solely evaluating their final outcomes.

We evaluate several state-of-the-art automatic research agents and LLMs on FML-bench. A central finding concerns agent strategy: once the basic requirements for both exploration breadth and depth are met, broader exploration proves more effective. Generating a wider variety of ideas more reliably leads to successful methods than repeatedly refining a single one, and we observe a positive correlation between idea diversity and performance improvement. Besides, we find that Gemini-2.5-Pro outperforms GPT-5 under our protocol. Finally, while CLI-style agents such as Claude Code offer general-purpose flexibility, they often fail to complete multi-step tasks due to early termination, where the model stops despite further actions being possible. This suggests that, although flexible, CLI-style agents are less suitable for automatic machine learning research than agents specifically designed for it.

We summarize our contributions as follows:

- We construct FML-bench, a benchmark centered on a diveristy of fundamental ML problems instantiated in real-world codebases, closing gaps left by usecase-oriented, engineering-heavy evaluations.

- We propose a five-dimensional evaluation protocol covering utility, diversity, academic contribution rate, cost, and step success rate, jointly measuring empirical progress, research quality, and reliability.

- We provide empirical insights on research strategy (breadth vs. depth of exploration), quantify the role of diversity in driving gains, and report comparative results across leading agent frameworks and LLMs, offering guidance for practical agent design.

## 2 RELATED WORKS

### 2.1 AUTOMATIC AI AGENTS

Recent advances in large language models (LLMs) have enabled research agents to support core components of the scientific workflow. These agents are capable of generating and prioritizing research ideas, retrieving and synthesizing literature, and simulating peer review processes. For instance, SciMON(Wang et al., 2024a) and Nova(Hu et al., 2024) implemented frameworks for generating diverse and novel research ideas. AutoSurvey(Wang et al., 2024b) presented an automated literature review framework that performs retrieval over a large arXiv corpus, followed by outline

planning and section drafting using specialized models. Meanwhile, AgentReview(Jin et al., 2024) employed LLM agents to simulate peer reviews, rebuttals, and committee discussions, offering insights into the dynamics of academic decision-making.

Recent efforts are moving beyond assistance toward fully automatic research agents. These systems aim not only to support researchers but to generate ideas, implement them, run experiments, and refine approaches without human supervision. One representative system is AIDE (Jiang et al., 2025), a tree-search agent that optimizes user-defined metrics by iteratively editing and evaluating code, though it executes only one file and modifies a specific target file per iteration. TheAIScientist (Lu et al., 2024) represents an independent line of work, demonstrating end-to-end autonomy across the research process, including idea generation, implementation, experimentation, analysis, and manuscript drafting. Its improved version (Yamada et al., 2025) further reduces reliance on hand-crafted templates, enhancing generality across tasks. Similarly, the AgentLaboratory executes a full pipeline for automatic research, but its evaluation is limited to relatively simple research questions. Separately, AlphaEvolve (Novikov et al., 2025) adopts an evolutionary approach, iteratively refining and selecting promising ideas through variation and empirical evaluation. Beyond the computer science domain, a growing number of research agents have been developed for other fields, including chemistry, where they are used to investigate and optimize chemical processes (Boiko et al., 2023; M. Bran et al., 2024), and biomedical science, where they have been applied to the discovery of novel nanobodies (Swanson et al., 2024).

## 2.2 BENCHMARKS FOR ML AGENTS

While recent benchmarks have begun to evaluate agents on code-intensive tasks, they remain limited in both scope and flexibility. MLAgentBench (Huang et al., 2023) includes 13 machine learning engineering tasks, but most are implemented as single-file scripts, which is not practical for real-world scenarios. In addition, it requires to set individual evaluator for each task and lacks support, limiting its scalability to support more tasks. MLE-Bench (Chan et al., 2024) covers 75 Kaggle competitions and assesses whether agents can function as machine learning engineers. It emphasizes tasks such as data pipeline management, experiment orchestration, and submission formatting, which may shift focus away from core machine learning understanding. ML-Dev-Bench (Padigela et al., 2025) places greater emphasis on engineering aspects such as dataset loading and API integration. It evaluates agents' ability to improve existing baselines only in performance tests, which are relatively simple due to narrow task scopes like classification and segmentation, and the use of fixed starter files. In contrast, our benchmark includes tasks spanning diverse machine learning domains. DSBench (Jing et al., 2024) aggregates 466 data analysis tasks and 74 modeling tasks from ModelOff and Kaggle, focusing on problem-solving within data science workflows. By comparison, our benchmark focuses on 8 diverse and fundamental machine learning research problems. It is built on real-world codebases, thereby providing practical challenges and strong extensibility by construction, while maintaining a low coding barrier.

## 3 UNIFIED EVALUATION FRAMEWORK FOR AUTOMATIC ML RESEARCH

Automatic ML research agents operate through iterative refinement cycles, where each iteration involves hypothesis generation, code modification, experimental execution, and empirical evaluation. To formalize this process, we propose a unified optimization framework that is explicitly aligned with five evaluation dimensions.

Consider an agent conducting research over $T$ iterations. At iteration $t \in \{1, \ldots, T\}$, the agent starts with codebase $\mathbf{C_{t-1}}$ and generates a hypothesis $h_t$ from a learned proposal distribution $q_t$. This hypothesis is instantiated as a concrete code modification $m_t$. After completing all $T$ iterations, the agent has produced a hypothesis set $\mathcal{H} = \{h_1, \ldots, h_T\}$ with corresponding modifications $\mathcal{M} = \{m_1, \ldots, m_T\}$. The agent's objective over the research process is:

$$\max_{T, \{q_t, m_t\}_{t=1}^T} \sum_{t=1}^{T} [U_t + \lambda A_t - \eta P_t] + \gamma S(\mathcal{M}, \mathbf{C_1}) + \beta D(\mathcal{H}) \tag{1}$$

where $U_t = \mathbb{E}_{h_t \sim q_t}[U(m_t, \mathbf{C_{t-1}})]$ is the expected utility, $A_t = A(m_t)$ is the academic contribution rate, $\mathbf{C_t} = \mathbf{C_{t-1}} \oplus m_t$, $P_t = P(m_t)$ is the cost, $S(\mathcal{M}, \mathbf{C_1})$ is the step success rate, and $D(\mathcal{H})$ measures diversity across all hypotheses.

**Evaluation metrics** Each term in Eq. 1 corresponds directly to our proposed evaluation metrics:

- **Utility** $U(m, \mathbf{C})$**:** The primary objective measuring empirical performance improvement. Specifically, $U(m, \mathbf{C}) = \text{perf}(\mathbf{C} \oplus m) - \text{perf}(\mathbf{C})$, where $\text{perf}(\cdot)$ evaluates the task-specific metric (e.g., accuracy, AUC, error rate).

- **Diversity** $D(\mathcal{H})$**:** Quantifies the variety across all proposed hypotheses $\mathcal{H} = \{h_1, \ldots, h_T\}$. We measure this through semantic and structural variance of the resulting modifications, capturing the agent's exploration breadth. Empirically, $D(\mathcal{H})$ strongly correlates with discovering high-performing solutions.

- **Academic Contribution Rate** $A(m)$**:** Measures the proportion of academic/algorithmic contributions (e.g., novel architectures, loss functions, training schemes) relative to engineering modifications (e.g., hyperparameter tuning, infrastructure fixes). Higher $A(m)$ indicates greater scientific contribution, distinguishing genuine research advances from implementation optimizations.

- **Step Success Rate** $S(\mathcal{M}, \mathbf{C_1})$**:** Captures the reliability of all code modifications $\mathcal{M}$ on initial codebase $\mathbf{C_1}$. This reflects the agent's ability to produce syntactically correct, semantically coherent code that successfully completes experiment iterations without errors.

- **Cost** $P(m)$**:** Encompasses time expenditure (wall-clock time) and API usage (tokens) required to execute codebase with modification $m$.

**Design Principles for Effective Agents.** To achieve high utility while maintaining research quality, effective agents should satisfy:

$$D(\mathcal{H}) \geq \delta, \quad \bar{A} \geq \alpha, \quad S \geq \rho, \quad \sum_{t=1}^{T} P_t \leq B \tag{2}$$

where $\bar{A} = \frac{1}{T}\sum_{t=1}^{T} A_t$ is the average academic contribution rate. These principles guide agent design: maintain exploration breadth to avoid local optima ($D \geq \delta$), prioritize algorithmic innovation ($\bar{A} \geq \alpha$), ensure reliable execution ($S \geq \rho$), and respect computational budgets ($\sum P_t \leq B$).

## 4 BENCHMARK DESIGN

### 4.1 TASK DESCRIPTION

We select 8 diverse tasks that collectively span the most critical aspects of ML research, ensuring comprehensive assessment for ML research agents. These tasks represent core competencies that a comprehensive and robust agent should demonstrate: the ability to generalize beyond training distributions, learn efficiently from limited data, discover meaningful representations, retain knowledge over time, reason about causal relationships, maintain reliability under adversarial conditions, protect sensitive information, and operate fairly across different groups.

Specifically, we evaluate agent performance across 8 critical machine learning tasks. 1) **Generalization.** Assessed via a cross-domain transfer task in which models train on a source domain and are evaluated on a held-out target under distribution shift. The objective is to maximize out-of-domain accuracy. 2) **Data Efficiency.** Tested through few-shot classification task. Agents should propose approaches to improve metric-based decision rules in the embedding space to boost accuracy with limited labels. 3) **Representation Learning.** Pretrain encoders in self-supervision manner and evaluate by linear-probe accuracy with the encoder frozen, targeting meaningful feature discovery. 4) **Continual Learning.** Measure knowledge retention in a class-incremental sequence with a shared output head. Agents should propose methods to mitigate catastrophic forgetting and maximize average accuracy across all tasks. 5) **Causality.** Estimate treatment effects under a specified causal data-generating process and minimize absolute error in the average treatment effect (ATE). 6) **Robustness and Reliability.** Evaluate resilience to adversarial corruption, including poisoning

or backdoor perturbations, while preserving clean performance. The defense score balances both objectives. 7) **Privacy.** Assess protection against information leakage by reducing the effectiveness of membership-inference attacks, i.e., lowering attack AUC. 8) **Fairness and Bias.** Evaluate equitable performance across groups in binary classification with sensitive attributes, aiming to improve group-fairness metrics (e.g., minimizing absolute average odds difference) without sacrificing overall accuracy.

## 4.2 UNIFIED INPUT-OUTPUT INTERFACE

A core design of our benchmark is to support direct utilization of machine learning research GitHub repositories. Real-world repositories vary significantly in execution pipelines, output formats, and evaluation protocols. We solve this through unified input-output interfaces that preserve repository complexity.

**Input** Existing benchmark designs struggle to handle different GitHub repositories. Data-based benchmarks accept only datasets and task descriptions as inputs. And benchmarks providing codebase assume unified training script names, single-stage training, no customizable arguments, or requiring to set a individual evaluator manually. In contrast, real repositories use different script names, multi-stage pipelines, diverse training arguments, and include their own evaluator already. Our solution treats the complete execution sequence (training and evaluation commands) as a single input unit so that the agent receives command list for running experiments. Therefore, our benchmark provide following resources(as shown in Fig. 1) to agent as input: 1) task description with objectives and expected outputs, 2) complete repository code, 3) suggested files for modification, 4) protected code segments that cannot be modified to preserve evaluation integrity, 5) command list for running experiments, 6) baseline performance, and 7) target improvement metrics.

**Output** Repository outputs are various (e.g. from text files to JSON formats). However, most outputs share a common structure: performance metrics on specific datasets. We provide a post-processing module that converts diverse task outputs into a standardized format, enabling consistent metric extraction across all tasks while preserving native output mechanisms. This design bridges the gap between evaluation standardization and real-world repository diversity, enabling rigorous assessment of agents on practical code optimization tasks.

**Evaluation Integrity** Our benchmark implements protective measures to ensure evaluation integrity. Specifically, we protect evaluation files as read-only and prevent agent modification. Agents must utilize these protected evaluation files to assess their proposed methods.

## 5 EXPERIMENTS

### 5.1 SETTINGS

**Selections of Agents** As shown in Fig. 2, we explore three automatic machine learning research agents, each adopting a distinct research strategy. TheAIScientist follows a broad exploration approach, generating and testing a wide range of hypotheses in parallel across multiple experimental directions. AIDE employs a hierarchical, tree-based search strategy, balancing the exploration of new possibilities with the exploitation of promising results. And we prompt Claude Code to employ a linear refinement strategy, sequentially improving its hypotheses and code implementations to address ML tasks.

**Benchmark Task Settings** Our benchmark encompasses 8 fundamental machine learning challenges, each implemented using established repositories with baselines: **1) Generalization** using the DomainBed (Gulrajani & Lopez-Paz, 2020) repository with ERM (Vapnik, 1998) baseline, optimizing out-of-domain accuracy; **2) Data Efficiency** using the Easy-Few-Shot-Learning (Sicara, 2024) repository with Prototypical Networks (Snell et al., 2017) baseline, maximizing few-shot classification accuracy; **3) Representation Learning** using the Lightly (Lightly-AI, 2025) repository with MoCo (He et al., 2020) baseline, maximizing linear probing accuracy on frozen encoders; **4) Continual Learning** using the Continual-Learning (van de Ven et al., 2022) repository with Synaptic Intelligence (Zenke et al., 2017) baseline, optimizing average accuracy across sequential

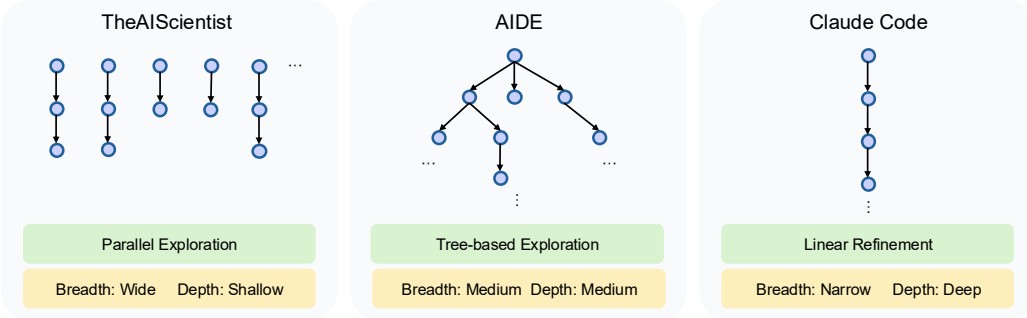

Figure 2: **Comparison of research exploration strategies of different agents.** TheAIScientist uses parallel exploration for broad coverage, AIDE employs hierarchical tree-based search balancing exploration and exploitation, while Claude Code follows linear refinement for sequential improvement.

tasks; **5) Causality** using the CausalML (Chen et al., 2020) repository with DragonNet (Shi et al., 2019) baseline, minimizing mean absolute error; **6) Robustness and Reliability** using the Adversarial Robustness Toolbox (ART) (Nicolae et al., 2018) repository with dp-instahide (Borgnia et al., 2021) defense baseline, optimizing defense scores against backdoor attacks; **7) Privacy** using the PrivacyMeter (Murakonda & Shokri, 2020) repository with Wide-ResNet-28-2 (Zagoruyko & Komodakis, 2016) baseline, minimizing membership inference attack AUC; and **8) Fairness and Bias** using the AIF360 (Bellamy et al., 2018) repository with Adversarial Debiasing baseline, minimizing absolute average odds difference while preserving classification performance. Comprehensive details are provided in Appendix B.

**Experimental Protocol**   Each agent is required to execute in three independent rounds. In each round, the agent is assigned a fixed budget of *total steps* = 100 (iterations). We select the best result achieved among the three rounds based on the target metric computed on the test set.

**Evaluation Metrics**   We evaluate agent performance on our benchmark using the proposed metrics: Utility, Diversity, Academic Contribution Rate, Cost, and Step Success Rate. In addition, we report the Step Completion Rate which calculate the proportion of executed steps relative to the required total, since AIDE and Claude Code may exhibit premature termination. See Sec. C in appendix for detailed calculation of metrics. We present detailed Utility results across the 8 tasks (Tab. 2), and report averages of the other metrics over the 8 tasks (Tab. 3); full per-task results are provided in Appendix F.

## 5.2   Implementation Details

**Agents**   To enable these agents to operate on our benchmark, several modifications were necessary. We adapted TheAIScientist by fixing compatibility issues and extending its functionality to support the requirements of our benchmark, such as executing experiments in real repositories and reporting results consistently. As for AIDE, we employed its cloud-based commercial variant, Weco, as the operational interface, adapting our benchmark to integrate with its workflow despite limited control over its internal mechanisms. For Claude Code, we designed a prompting scheme (see Appendix G) that enabled it to function as an automatic research agent, capable of reading code, generating hypotheses, and proposing modifications grounded in experimental feedback.

**LLMs adopted for agents**   We employ GPT-5 (2025-08-07) and Gemini-2.5-Pro (2025-06-17) for TheAIScientist and AIDE. For Claude Code, it is constrained to its native models and therefore we use Opus-4.1 (2025-08-05).

## 5.3   Results and Discoveries

### 5.3.1   Comparison of Agents with Different Research Strategies

As shown in Tab. 2, the combination of TheAIScientist with Gemini-2.5-Pro achieved the best performance, ranking first in 4 out of 8 tasks. The combination of AIDE with Gemini-2.5-Pro ranked

Table 2: **Comparison of best performance (Utility) among different agents.** G2.5-Pro denotes Gemini-2.5-Pro. (↑) indicates higher is better, (↓) indicates lower is better.

| ML Problems | Baseline | The AI Scientist | | AIDE | | Claude Code |
|---|---|---|---|---|---|---|
| | | GPT-5 | G2.5-Pro | GPT-5 | G2.5-Pro | Opus-4.1 |
| Generalization (↑) | 0.2254 | **0.5036** | 0.3252 | 0.2254 | 0.2254 | **0.5036** |
| Data Eff. (↑) | 0.6547 | 0.7689 | **0.8231** | 0.6547 | 0.6547 | 0.6571 |
| Rep. Learn. (↑) | 0.7562 | 0.7796 | **0.8597** | 0.8469 | 0.8466 | 0.7725 |
| Cont. Learn. (↑) | 0.2710 | 0.4281 | **0.7808** | 0.4369 | 0.3658 | 0.2337 |
| Causality (↓) | 1.1445 | 1.0063 | 0.9925 | 0.9683 | **0.9549** | 0.9840 |
| Robust. & Rel.(↑) | 0.4848 | 0.9311 | 0.9205 | 0.9174 | **0.9633** | 0.8921 |
| Privacy (↓) | 0.8114 | 0.4908 | **0.1750** | 0.4882 | 0.4814 | 0.4892 |
| Fair. & Bias (↓) | 0.3787 | 0.0603 | 0.1002 | **0.0385** | 0.0917 | 0.3787 |

Table 3: **Comparisons of different agents across diversity, academic contribution rate, cost, and success metrics.** Contrib. denotes Contribution and Cons. stands for Consumption.

| Metrics | The AI Scientist | | AIDE | | Claude Code |
|---|---|---|---|---|---|
| | GPT-5 | G2.5-Pro | GPT-5 | G2.5-Pro | Opus-4.1 |
| Diversity | $28.46^{\pm 9.97}$ | $20.86^{\pm 4.42}$ | $28.60^{\pm 13.32}$ | $18.41^{\pm 12.68}$ | $8.75^{\pm 6.07}$ |
| Academic Contrib. Rate | $0.89^{\pm 0.15}$ | $0.73^{\pm 0.12}$ | $0.79^{\pm 0.18}$ | $0.62^{\pm 0.09}$ | $0.51^{\pm 0.14}$ |
| Total Token Cons. (M) (↓) | $6.07^{\pm 2.05}$ | $5.33^{\pm 2.55}$ | $\mathbf{0.85}^{\pm 0.59}$ | $2.59^{\pm 0.90}$ | $8.32^{\pm 3.10}$ |
| Step Token Cons. (M) (↓) | $0.06^{\pm 0.02}$ | $0.05^{\pm 0.03}$ | $\mathbf{0.03}^{\pm 0.03}$ | $0.04^{\pm 0.03}$ | $1.77^{\pm 1.32}$ |
| Total Time Cons. (H) (↓) | $8.32^{\pm 3.15}$ | $12.23^{\pm 7.93}$ | $2.46^{\pm 1.77}$ | $8.06^{\pm 6.44}$ | $\mathbf{1.27}^{\pm 1.77}$ |
| Step Time Cons. (Min) (↓) | $10.39^{\pm 10.79}$ | $12.73^{\pm 14.64}$ | $10.79^{\pm 12.86}$ | $\mathbf{8.65}^{\pm 7.92}$ | $12.49^{\pm 10.77}$ |
| Step Success Rate (↑) | $\mathbf{0.92}^{\pm 0.08}$ | $0.87^{\pm 0.12}$ | $0.65^{\pm 0.25}$ | $0.73^{\pm 0.22}$ | $0.83^{\pm 0.18}$ |
| Step Completion Rate (↑) | $\mathbf{1.00}^{\pm 0.00}$ | $\mathbf{1.00}^{\pm 0.00}$ | $0.38^{\pm 0.22}$ | $0.79^{\pm 0.25}$ | $0.07^{\pm 0.05}$ |

second, securing top results in 2 out of the 8 tasks. These findings suggest that TheAIScientist performs better in discovering novel and effective machine learning methods, compared to AIDE and Claude Code.

As illustrated in Fig. 2, TheAIScientist adopts a research exploration strategy that is broad but shallow, while AIDE maintains both medium breadth and depth. In contrast, Claude Code exhibits a narrow yet deep exploration pattern. For a detailed explanation of the research strategies adopted by each agent, refer to Section 5.1. Considering the research exploration strategy together, the results suggest that a broader research exploration space adopted by TheAIScientist is more effective for discovering promising ideas. This insight offers practical guidance for real-world research: **broadly exploring diverse ideas could be more productive than focusing on a single direction**.

### 5.3.2 ANALYSIS OF DIVERSITY

In the calculation of Diversity, we employ GraphCodeBERT (Guo et al., 2020) to compute embeddings of each step's code. GraphCodeBERT leverages data-flow graphs and code structure rather than surface tokens alone. Consequently, code snippets with identical logic but different variable/function names receive high similarity scores due to their shared structural patterns. Such pairs will be treated as low diversity in our metric.

As shown in Tab. 2 and Tab. 3, several key findings emerge from our experimental results. First, TheAIScientist and AIDE demonstrate significantly higher diversity compared to Claude Code, while also exhibiting superior overall performance across the tasks. Second, when both systems employ GPT-5 as the underlying language model, TheAIScientist and AIDE achieve comparable diversity scores (28.46 vs. 28.60, respectively), and both methods attain one best result each. Third, when utilizing Gemini-2.5-Pro, TheAIScientist exhibits higher diversity than AIDE (20.85 vs. 18.41) and achieves a greater number of best results (4 vs. 2). These observations collectively suggest a positive correlation between exploration diversity and outcome performance, suggesting that higher diversity tends to yield better results.

Table 4: **Correlation analysis between diversity and outcome.** When computing the correlation, the outcome scores are negated for tasks where lower values indicate better performance, ensuring that positive correlations consistently represent improved performance with increased diversity.

| Metric | Gen. | Data Eff. | Rep. Learn. | Cont. Learn. | Causality | Robust. | Privacy | Fair. |
|---|---|---|---|---|---|---|---|---|
| p-value | 0.1137 | 0.0121 | 0.1358 | 0.0002 | 0.4668 | 0.1944 | 0.6479 | 0.0517 |
| Correlation | **0.4418** | **0.6287** | **0.4036** | **0.8374** | -0.2036 | -0.3846 | -0.1286 | **0.5292** |

Furthermore, TheAIScientist achieves a mean diversity of 24.66, which exceeds both AIDE's 23.50 and Claude Code's 8.75. This enhanced diversity can potentially be attributed to TheAIScientist's more exploration breadth-oriented search strategy, as illustrated in Sec. 5.3.1.

To further validate the relationship between diversity and performance, we computed the correlation coefficients between exploration diversity and outcome performance. As presented in Tab. 4, our findings reveal positive correlations across five tasks, with p-values indicating significant, borderline significant, or suggestive evidence of this relationship ($p < 0.05$ for significant; $p < 0.10$ for borderline significant; $p < 0.15$ for suggestive evidence; $p \geq 0.15$ for not significant). Notably, for these five tasks, higher exploration diversity was associated with better outcomes.

### 5.3.3 ACADEMIC CONTRIBUTION RATE

The academic contribution rate provides further insights into the characteristics of each agent. It helps disentangle the impact of academic value from other factors such as engineering effort and diversity. This metric is designed to encourage agents to propose new ML methods that differ from the baseline rather than merely tuning hyperparameters (engineering modification).

Specifically, academic contribution rate metric compares each step with the baseline method rather than comparing consecutive steps with each other. Any method that introduces a new machine learning idea (e.g. a novel objective or inductive bias) is classified as an academic modification, regardless of its complexity. For example, if $step_k$ proposes a new method and is classified as an academic step, then $step_{k+1}$, which refines $step_k$, is also classified as an academic step because both are compared against the baseline rather than against each other. Therefore, both initial methodological proposals and subsequent refinement steps are classified as academic contributions.

We conducted a human evaluation to compute academic contribution ratio. Three human annotators each evaluated 200 samples (5 agents × 40 samples per agent). As shown in Tab. 3, TheAIScientist generally exhibits a higher academic contribution rate than AIDE, whereas Claude Code consistently shows the lowest rate. This suggests that the ideas and code modifications proposed by TheAIScientist are more closely aligned with methodological advancements, rather than relying on engineering to boost performance. Furthermore, comparing GPT-5 and Gemini-2.5-Pro reveals that Gemini-2.5-Pro tends to propose more engineering-oriented solutions than GPT-5, but Gemini-2.5-Pro demonstrates superior performance compared to GPT-5 (as shown in Tab. 2). These results suggest that both the capacity to propose academic contribution and engineering capabilities influence the outcomes, indicating that strong abilities in both aspects are necessary.

### 5.3.4 TOKEN AND TIME CONSUMPTION

Tab. 3 reports both the average token and time consumption per step, as well as the total token and time usage for a complete experimental run. We observe that TheAIScientist consumes more tokens than AIDE, while Claude Code, despite its lower performance, uses the highest number of tokens among the three agents. This indicates that dedicated automatic ML research agents, such as TheAIScientist and AIDE, are more suitable for ML research problems in terms of both performance and token efficiency compared to general-purpose agents like Claude Code. In terms of time per step, all three agents show similar durations, with differences of around 2 minutes, which are not substantial. However, AIDE and Claude Code exhibit significantly shorter total execution times for a full experiment. This is primarily due to premature termination issues, as evidenced by the step completion rate, which leads to reduced overall time usage.

### 5.3.5 CASE ANALYSIS

**Worst Case Analysis** TheAIScientist strongly favors algorithmic modifications (56.2%) with zero implementation errors; AIDE presents balanced distribution but higher implementation issues (31.2%); while Claude Code shows elevated parameter configuration rates (37.5%) matching its algorithmic modification rate. Across LLMs, Gemini-2.5-Pro exhibits extremely strong algorithmic modification tendency (68.8%), GPT-5 primarily focuses on structural modifications (50.0%), and Opus-4.1 maintains balanced distribution between parameter configuration and algorithmic modifications. See Appendix D for detailed results.

**Best Case Analysis** TheAIScientist shows strong algorithmic innovation focus, predominantly targeting core algorithmic logic; AIDE combines strong algorithmic innovation with hyperparameter tuning; while Claude Code uniquely favors architectural modifications including layer additions and normalization adjustments. Across LLMs, GPT-5 exclusively prioritizes algorithmic modifications (100.0%), Gemini-2.5-Pro exhibits the most diversified approach with unique data augmentation efforts (28.6%), and Opus-4.1 balances attention between architectural and algorithmic modifications. See Appendix D for detailed results.

### 5.3.6 ADDITIONAL OBSERVATIONS

**Characteristics of Claude Code** Since all actions are executed based on LLM decisions rather than fixed procedures, Claude Code often fails to follow prompt instructions, frequently terminating experiments prematurely. Despite this, it demonstrates a high improvement speed (refer to the improvement curves Fig. 3 and speed comparison Tab. 16 in the Appendix). In addition, its academic contribution rate is low, with a strong emphasis on engineering. This may be attributed to its nature as a general-purpose agent, rather than a specialized automatic ML research agent.

**Shallow edits** We found that AIDE sometimes misinterprets the structure and logic of target codebases. In certain cases, it generated new classes or components that were never integrated into the actual execution pipeline, resulting in no functional improvement over the baseline. As shown in Tab. 2, AIDE failed to improve the baseline in tasks related to Generalization and Data Efficiency. This may stem from the fact that AIDE only supports iterative modifications on a single file. However, real-world ML research codebases are often complex and span multiple files, making AIDE insufficient for addressing realistic research tasks.

**Premature termination** We encountered the issue of early termination of AIDE and Claude Code. For AIDE, the agent sometimes terminated prematurely due to its commercial version Weco which relies on cloud infrastructure that occasionally failed during execution. For Claude Code, early stopping was often triggered by the model's internal reasoning, where the LLM would decide not to continue even when further actions are possible.

## 6 CONCLUSION

In this work, we introduce FML-bench, a benchmark that assesses automatic machine learning research agents on 8 diverse and fundamental ML problems drawn from real-world codebases and authentic research workflows. Within the benchmark, we introduce a five-dimensional evaluation protocol that facilitates a more systematic assessment, moving beyond exclusive reliance on final performance. Building on FML-bench, we conducted a systematic analysis of three state-of-the-art automatic research agents. Our findings reveal that agents capable of generating and evaluating multiple hypotheses across diverse directions, such as TheAIScientist, tend to outperform those that focus on iteratively refining a single line of thought (e.g., Claude Code). These findings suggest that a broader exploratory capacity contributes more significantly to overall success once a basic level of proficiency is established. Overall, FML-bench provides a robust and practical foundation for evaluating the capabilities of research agents and offers a pathway toward building more effective, generalizable, and scientifically productive research agents.

## REPRODUCIBILITY STATEMENT

We prioritize reproducibility and provide all benchmark code, experiment prompts, and configuration files in anonymous github: `https://anonymous.4open.science/r/Anonymous-78B6`. The paper describes the benchmark design, evaluation protocol, experimental settings, and implementation details, including baseline configurations and prompt specifications.

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

APPENDIX

## A    USE OF LLMS

In this study, we employed LLMs (ChatGPT and Claude AI) to assist with manuscript writing. Specifically, these tools were used to polish the writing (e.g., improving fluency, grammar, and clarity of expression), aid in formatting, and support proofreading. Additionally, we used them to facilitate the retrieval of related work by suggesting potentially relevant literature.

## B    TASK SETTINGS

**Generalization**    We adopt the *DomainBed* (Gulrajani & Lopez-Paz, 2020) repository with *Empirical Risk Minimization (ERM)* (Vapnik, 1998) as the baseline and evaluate on the *ColorMNIST* (Arjovsky et al., 2019) dataset. The evaluation metric is accuracy on a held-out domain. The agent is required to train on a source domain and generalize to a target domain under distribution shift. The task objective is to improve out-of-domain generalization performance. This task evaluates the agent's ability to develop algorithms that transfer effectively across domains, which is critical for robust real-world deployment.

**Causality**    We utilize the *CausalML* (Chen et al., 2020) repository with *DragonNet* (Shi et al., 2019) as the baseline model, evaluated on IHDP dataset (Hill, 2011). The performance metric is mean absolute error (MAE). The agent's goal is to develop improved causal inference strategies that minimize this error. This task assesses the agent's capacity to reason about interventions, which is essential for decision-making in high-stakes environments.

**Data Efficiency**    For data-efficient learning, we employ the *Easy-Few-Shot-Learning* (Sicara, 2024) repository with *Prototypical Networks* (Snell et al., 2017) as the baseline, evaluated on the *Mini-ImageNet* (Vinyals et al., 2016) dataset. The agent must operate under a frozen backbone and propose improved algorithms for metric-based classification in the embedding space. Accuracy is used as the evaluation metric. This task measures the agent's ability to enhance few-shot learning performance by optimizing distance-based reasoning under tight data constraints.

**Robustness and Reliability**    We adopt the *Adversarial Robustness Toolbox (ART)* (Nicolae et al., 2018) repository, using *dp-instahide* (Borgnia et al., 2021) as the defense baseline on a poisoned *MNIST* (Deng, 2012) dataset constructed with backdoor method including the edge-based triggers and various distributed attack patterns. The evaluation metric is a defense score, defined as the harmonic mean of clean accuracy and resistance accuracy against backdoor attacks. The agent is tasked with proposing defenses that reduce the effectiveness of poisoning attacks while maintaining high clean performance. This task probes the agent's ability to improve model robustness against adversarial corruption.

**Privacy**    This task uses the *PrivacyMeter* (Murakonda & Shokri, 2020) repository and *Wide-ResNet-28-2* (Zagoruyko & Komodakis, 2016) as the baseline model trained on *CIFAR-10* (Krizhevsky et al., 2009). The evaluation targets membership inference attacks, with performance measured by the area under the ROC curve (AUC). The agent must design improved defense mechanisms that minimize the AUC, thereby reducing information leakage. This task is critical for assessing the agent's ability to enhance privacy protections in the face of inference attacks.

**Representation Learning**    For self-supervised learning, we employ the *Lightly* (Lightly-AI, 2025) repository with *MoCo* (He et al., 2020) as the baseline and evaluate on *CIFAR-10* (Krizhevsky et al., 2009). The primary evaluation metric is the accuracy of a linear classifier trained on top of the frozen encoder after pretraining. The agent is expected to devise improved representation learning methods that yield higher linear probing accuracy. This task tests the agent's ability to learn generalizable and semantically meaningful features from unlabeled data.

**Continual Learning**    We use the *Continual-Learning* (van de Ven et al., 2022) repository with *Synaptic Intelligence (SI)* (Zenke et al., 2017) as the baseline, evaluated on *splitMNIST* (Deng, 2012)

in the class-incremental learning (Class-IL) scenario. The model must learn sequentially across multiple contexts using a shared 10-way classification head. The evaluation metric is the average accuracy across all tasks. The agent is expected to develop algorithms that mitigate catastrophic forgetting and maintain performance across contexts. This task evaluates long-term adaptability in non-stationary environments.

**Fairness and Bias** We adopt the *AIF360* (Bellamy et al., 2018) repository with *Adversarial Debiasing* as the baseline, using the *COMPAS* dataset (Angwin et al., 2016). The primary fairness metric is average odds difference and the agent is tasked with enhancing fairness (minimizing absolute average odds difference) while maintaining or improving classification accuracy. This task measures the agent's ability to balance equitable outcomes with model utility across demographic groups.

## C    EVALUATION METRICS

**Utility** reflects empirical improvement within each task and is reported using the task-specific performance metric.

**Diversity** quantifies implementation dispersion within the best-performing round. Given $i$ th step code embedding $\mathbf{e}_i$ extracted from GraphCodeBERT (Guo et al., 2020) and centroid $\bar{\mathbf{e}} = \frac{1}{n}\sum_{i=1}^{n}\mathbf{e}_i$, diversity is computed as $\frac{1}{n}\sum_{i=1}^{n}\|\mathbf{e}_i - \bar{\mathbf{e}}\|_2$. Greater dispersion indicates broader exploration of implementation choices within a single round.

**Academic Contribution Rate** reflects the agent's tendency toward innovative algorithmic contributions versus conventional engineering optimizations. It is defined as $N_{\text{aca}}/N_{\text{suc}}$, where $N_{\text{aca}}$ is the number of steps recogniezd by Qwen3 as academic modifications (e.g., new losses or architectures) relative to the baseline, and $N_{\text{suc}}$ is the number of steps whose experiments execute without errors and yield valid results.

**Step Success Rate** is defined as the $N_{\text{suc}}/N_{\text{comp}}$. Combining the metric Step Completion Rate ($N_{\text{comp}}/N_{\text{total}}$) will reflect the system reliability of the agent. Here, $N_{\text{comp}}$ is the number of steps actually executed and $N_{\text{total}}$ is the number of steps assigned to the agent.

**Cost** is reported as total token consumption (sum across steps) and step token consumption (average per step), together with total time consumption and step time consumption, to characterize computational and temporal efficiency.

## D    CASE ANALYSIS

### D.1    WORST CASE ANALYSIS

**Modification Distribution by Agents** As shown in Tab. 5, TheAIScientist strongly favors algorithmic modifications (56.2%) with zero implementation errors, indicating robust execution. AIDE presents a more balanced modification distribution but experiences higher implementation and execution issues (31.2%). Claude Code shows elevated parameter configuration rates (37.5%), equivalent to its algorithmic modification rate.

**Modification Distribution by LLMs** As shown in Tab. 6, Gemini-2.5-Pro strongly favors algorithmic modifications (68.8%), preferring core algorithm changes; GPT-5 primarily focuses on structural modifications (50.0%) with emphasis on architecture adjustments; while Opus-4.1 maintains balanced distribution between parameter configuration and algorithmic modifications.

### D.2    BEST CASE ANALYSIS

**Modification Distribution by Agents** As shown in Tab. 7, TheAIScientist shows strong algorithmic innovation focus, predominantly targeting core algorithmic logic; AIDE combines strong algorithmic innovation with hyperparameter tuning; while Claude Code uniquely favors architectural modifications, including layer additions and normalization adjustments.

**Modification Distribution by LLMs**  As shown in Tab. 8, GPT-5 exclusively prioritizes algorithmic modifications (100.0%); Gemini-2.5-Pro exhibits the most diversified approach, uniquely incorporating substantial data augmentation efforts (28.6%); while Opus-4.1 balances attention between architectural and algorithmic modifications.

Table 5: Worst Case Modification Distribution by Agents

| Modification Type | The AI Scientist | AIDE | Claude Code |
|---|---|---|---|
| Algorithmic Modification | 56.2% (9) | 37.5% (6) | 37.5% (3) |
| Structural Modification | 31.2% (5) | 25.0% (4) | 12.5% (1) |
| Parameter Configuration | 12.5% (2) | 6.2% (1) | 37.5% (3) |
| Implementation & Execution | 0.0% (0) | 31.2% (5) | 12.5% (1) |

Table 6: Worst Case Modification Distribution by LLMs

| Modification Type | GPT-5 | Gemini-2.5-pro | Opus-4.1 |
|---|---|---|---|
| Algorithmic Modification | 25.0% (4) | 68.8% (11) | 37.5% (3) |
| Structural Modification | 50.0% (8) | 6.2% (1) | 12.5% (1) |
| Parameter Configuration | 6.2% (1) | 12.5% (2) | 37.5% (3) |
| Implementation & Execution | 18.8% (3) | 12.5% (2) | 12.5% (1) |

Table 7: Best Case Modification Distribution by Agents

| Modification Type | The AI Scientist | AIDE | Claude Code |
|---|---|---|---|
| Algorithmic Modification | 75.0% (12) | 83.3% (10) | 37.5% (3) |
| Structural Modification | 12.5% (2) | 0.0% (0) | 37.5% (3) |
| Data Processing | 12.5% (2) | 16.7% (2) | 0.0% (0) |
| Parameter Configuration | 0.0% (0) | 0.0% (0) | 25.0% (2) |

Table 8: Best Case Modification Distribution by LLMs

| Modification Type | GPT-5 | Gemini-2.5-pro | Opus-4.1 |
|---|---|---|---|
| Algorithmic Modification | 14 (100.0%) | 8 (57.1%) | 3 (37.5%) |
| Structural Modification | 0 (0.0%) | 2 (14.3%) | 3 (37.5%) |
| Data Processing | 0 (0.0%) | 4 (28.6%) | 0 (0.0%) |
| Parameter Configuration | 0 (0.0%) | 0 (0.0%) | 2 (25.0%) |

# E    PROTECTING EVALUATION INTEGRITY

A crucial aspect of our implementation involved protecting evaluation files from inadvertent modification by the agents. We employed agent-specific protection strategies: For TheAIScientist, we explicitly instructed the agent through prompting to avoid modifying evaluation files and implemented a systematic refresh mechanism that restores evaluation files before each evaluation cycle. For AIDE, protection is inherently ensured by its single-file modification constraint, which guarantees that when the target code and evaluation code are separated into different files, the evaluation files remain naturally protected. For Claude Code, we implemented a two-pronged approach in which the evaluation files were first restricted to read and execute permissions before running the agent, and then the `--disallowedTools` argument was employed to explicitly prevent permission modification operations during execution.

# F ADDITIONAL RESULTS

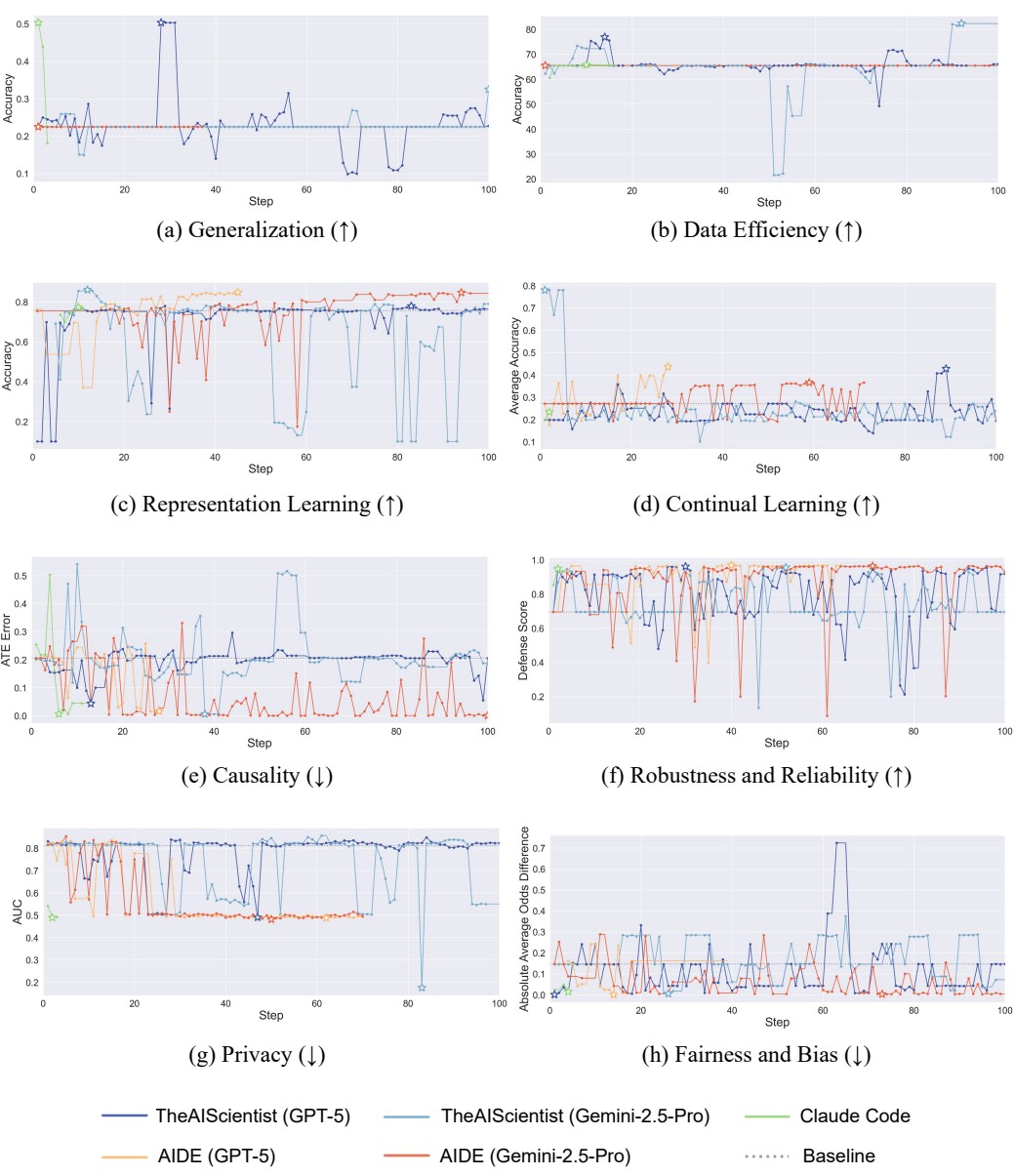

Figure 3: Agents' performance improvement curves across 8 tasks.

This section provides a comprehensive comparative analysis of automated AI research systems, focusing on both their performance and operational efficiency. We evaluate The AI Scientist, AIDE, and Claude Code across eight core machine learning research problems, considering not only their final outcomes but also their operational behavior over time.

Additional analysis results covers multiple key dimensions: detailed exploration diversity, computational efficiency in terms of total token usage (Tab. 10) and step-wise token usage (Tab. 11), runtime efficiency in terms of total time (Tab. 12) and step-wise time (Tab. 13), operational reliability measured by step success rate (Tab. 14) and step completion rate (Tab. 15), and improvement speed (Tab. 16). Together, these results provide detailed insights into the trade-offs between different agentic research designs, illustrating how they influence both the quality of research outcomes and the efficiency of resource utilization.

Table 9: **Code Diversity Comparison of Best Performance Round.** G2.5-Pro denotes Gemini-2.5-Pro. Bold numbers indicate the best (highest) performance in each row.

| ML Problems | The AI Scientist* | | AIDE* | | Claude Code |
|---|---|---|---|---|---|
| | GPT-5 | G2.5-Pro | GPT-5 | G2.5-Pro | Opus-4.1 |
| Generalization | 39.32 | 18.60 | 4.36 | 48.23 | 19.50 |
| Data Eff. | 28.18 | 20.19 | 18.03 | 10.92 | 10.08 |
| Rep. Learn. | 15.28 | 11.55 | 18.91 | 9.71 | 5.30 |
| Cont. Learn. | 18.86 | 24.83 | 36.72 | 18.80 | 2.71 |
| Causality | 24.03 | 21.37 | 38.61 | 11.65 | 6.43 |
| Robust. & Rel. | 45.44 | 24.04 | 43.03 | 21.23 | 16.19 |
| Privacy | 29.66 | 21.17 | 33.16 | 13.67 | 5.97 |
| Fair. & Bias | 26.88 | 25.14 | 35.97 | 13.10 | 3.84 |
| Average $\pm$ Std | $28.46^{\pm 9.97}$ | $20.86^{\pm 4.42}$ | $28.60^{\pm 13.32}$ | $18.41^{\pm 12.68}$ | $8.75^{\pm 6.07}$ |

Table 10: **Total Token Consumption Comparison (in millions).** G2.5-Pro denotes Gemini-2.5-Pro. Bold numbers indicate the best (lowest) performance in each row.

| ML Problems | The AI Scientist* | | AIDE* | | Claude Code |
|---|---|---|---|---|---|
| | GPT-5 | G2.5-Pro | GPT-5 | G2.5-Pro | Opus-4.1 |
| Generalization | 9.34 | 9.88 | **0.18** | 3.80 | 5.98 |
| Data Eff. | 3.09 | 2.55 | **0.33** | 1.53 | 7.98 |
| Rep. Learn. | 4.69 | 3.99 | **0.40** | 1.51 | 8.89 |
| Cont. Learn. | 7.81 | 7.74 | **1.06** | 3.21 | 2.99 |
| Causality | 4.57 | 3.05 | **0.56** | 2.99 | 9.26 |
| Robust. & Rel. | 7.24 | 6.74 | **1.69** | 2.93 | 8.22 |
| Privacy | 6.64 | 4.45 | 1.68 | **1.61** | 13.88 |
| Fair. & Bias | 5.13 | 4.21 | **0.94** | 3.12 | 9.34 |
| Average $\pm$ Std | $6.07^{\pm 2.05}$ | $5.33^{\pm 2.55}$ | $0.85^{\pm 0.59}$ | $2.59^{\pm 0.90}$ | $8.32^{\pm 3.10}$ |

Table 11: **Step Token Consumption Comparison (in millions).** G2.5-Pro denotes Gemini-2.5-Pro. Bold numbers indicate the best (lowest) performance in each row.

| ML Problems | The AI Scientist* | | AIDE* | | Claude Code |
|---|---|---|---|---|---|
| | GPT-5 | G2.5-Pro | GPT-5 | G2.5-Pro | Opus-4.1 |
| Generalization | **0.09** | 0.10 | **0.09** | 0.10 | 1.99 |
| Data Eff. | 0.03 | 0.03 | **0.01** | 0.02 | 0.50 |
| Rep. Learn. | 0.05 | 0.04 | **0.01** | **0.01** | 0.81 |
| Cont. Learn. | 0.08 | 0.08 | **0.04** | 0.05 | 1.49 |
| Causality | 0.05 | 0.03 | **0.02** | 0.03 | 0.77 |
| Robust. & Rel. | 0.07 | 0.07 | **0.03** | 0.05 | 1.64 |
| Privacy | 0.07 | 0.04 | **0.02** | **0.02** | 4.63 |
| Fair. & Bias | 0.05 | 0.04 | **0.02** | 0.03 | 2.34 |
| Average $\pm$ Std | $0.06^{\pm 0.02}$ | $0.05^{\pm 0.03}$ | $\mathbf{0.03}^{\pm 0.03}$ | $0.04^{\pm 0.03}$ | $1.77^{\pm 1.32}$ |

Table 12: **Total Time Consumption Comparison (in hours).** G2.5-Pro denotes Gemini-2.5-Pro. Bold numbers indicate the best (lowest) performance in each row.

| ML Problems | The AI Scientist* | | AIDE* | | Claude Code |
|---|---|---|---|---|---|
| | GPT-5 | G2.5-Pro | GPT-5 | G2.5-Pro | Opus-4.1 |
| Generalization | 9.91 | 17.42 | **0.20** | 6.07 | 0.81 |
| Data Eff. | 4.47 | 4.33 | 1.02 | 2.40 | **0.47** |
| Rep. Learn. | 7.50 | 23.96 | 4.45 | 14.49 | **5.56** |
| Cont. Learn. | 14.68 | 15.17 | 3.70 | 7.08 | **0.28** |
| Causality | 5.41 | 4.85 | 1.44 | 2.91 | **0.57** |
| Robust. & Rel. | 8.26 | 8.65 | 5.10 | 6.00 | **0.90** |
| Privacy | 9.24 | 20.03 | 2.43 | 21.04 | **1.29** |
| Fair. & Bias | 7.05 | 3.46 | 1.32 | 4.46 | **0.25** |
| Average $\pm$ Std | $8.32^{\pm3.15}$ | $12.23^{\pm7.93}$ | $2.46^{\pm1.77}$ | $8.06^{\pm6.44}$ | $\mathbf{1.27}^{\pm1.77}$ |

Table 13: **Step Time Consumption Comparison (in minutes).** G2.5-Pro denotes Gemini-2.5-Pro. Bold numbers indicate the best (lowest) performance in each row.

| ML Problems | The AI Scientist* | | AIDE* | | Claude Code |
|---|---|---|---|---|---|
| | GPT-5 | G2.5-Pro | GPT-5 | G2.5-Pro | Opus-4.1 |
| Generalization | **5.95** | 10.43 | 5.97 | 9.83 | 16.13 |
| Data Eff. | 2.68 | 2.60 | 2.03 | **1.43** | 1.77 |
| Rep. Learn. | 33.28 | 43.15 | 37.92 | **22.87** | 30.30 |
| Cont. Learn. | 8.80 | 9.08 | 7.93 | **5.98** | 8.33 |
| Causality | 3.25 | 2.90 | 3.08 | **1.73** | 2.87 |
| Robust. & Rel. | 4.95 | 5.20 | **4.77** | 6.67 | 10.83 |
| Privacy | 19.93 | 26.42 | 22.63 | **18.03** | 25.87 |
| Fair. & Bias | 4.23 | 2.07 | **2.02** | 2.63 | 3.80 |
| Average $\pm$ Std | $10.39^{\pm10.79}$ | $12.73^{\pm14.64}$ | $10.79^{\pm12.86}$ | $\mathbf{8.65}^{\pm7.92}$ | $12.49^{\pm10.77}$ |

Table 14: **Step Success Rate Comparison.** G2.5-Pro denotes Gemini-2.5-Pro. Bold numbers indicate the best (highest) performance in each row.

| ML Problems | The AI Scientist* | | AIDE* | | Claude Code |
|---|---|---|---|---|---|
| | GPT-5 | G2.5-Pro | GPT-5 | G2.5-Pro | Opus-4.1 |
| Generalization | 0.7700 | 0.9900 | 0.5000 | 0.7027 | **1.0000** |
| Data Eff. | **0.9600** | 0.6500 | 0.2333 | 0.2871 | 0.9375 |
| Rep. Learn. | 0.8300 | **0.8400** | 0.8000 | 0.6040 | 0.5455 |
| Cont. Learn. | 0.9600 | 0.9400 | 0.8214 | 0.9155 | **1.0000** |
| Causality | 0.8900 | 0.9000 | 0.8214 | **0.9406** | 0.8333 |
| Robust. & Rel. | **0.9900** | 0.9500 | 0.8281 | 0.6667 | 0.6000 |
| Privacy | 0.9700 | 0.9500 | 0.8714 | 0.9571 | **1.0000** |
| Fair. & Bias | **0.9500** | 0.7200 | 0.3590 | 0.7426 | 0.7500 |
| Average $\pm$ Std | $\mathbf{0.92}^{\pm0.08}$ | $0.87^{\pm0.12}$ | $0.65^{\pm0.25}$ | $0.73^{\pm0.22}$ | $0.83^{\pm0.18}$ |

Table 15: **Step Completion Rate (SCR) Comparison.** G2.5-Pro denotes Gemini-2.5-Pro. Bold numbers indicate the best (highest) performance in each row.

| ML Problems | The AI Scientist* | | AIDE* | | Claude Code |
|---|---|---|---|---|---|
| | GPT-5 | G2.5-Pro | GPT-5 | G2.5-Pro | Opus-4.1 |
| Generalization | **1** | **1** | 0.02 | 0.36 | 0.03 |
| Data Eff. | **1** | **1** | 0.30 | **1** | 0.16 |
| Rep. Learn. | **1** | **1** | 0.45 | **1** | 0.11 |
| Cont. Learn. | **1** | **1** | 0.28 | 0.70 | 0.02 |
| Causality | **1** | **1** | 0.28 | **1** | 0.12 |
| Robust. & Rel. | **1** | **1** | 0.64 | 0.53 | 0.05 |
| Privacy | **1** | **1** | 0.70 | 0.69 | 0.03 |
| Fair. & Bias | **1** | **1** | 0.39 | **1** | 0.04 |
| Average $\pm$ Std | **1.00**$^{\pm 0.00}$ | **1.00**$^{\pm 0.00}$ | 0.38$^{\pm 0.22}$ | 0.79$^{\pm 0.25}$ | 0.07$^{\pm 0.05}$ |

Table 16: **Improvement Speed Comparison.** We report the number of steps used to reach the threshold, which is defined as the best result achieved in the best run of the worst-performing agent. Less steps indicate higher improvement speed. G2.5-Pro denotes Gemini-2.5-Pro. Bold numbers indicate the best (lowest) performance in each row.

| ML Problems | The AI Scientist* | | AIDE* | | Claude Code |
|---|---|---|---|---|---|
| | GPT-5 | G2.5-Pro | GPT-5 | G2.5-Pro | Opus-4.1 |
| Generalization | **1** | **1** | **1** | **1** | **1** |
| Data Eff. | **1** | 2 | **1** | **1** | 3 |
| Rep. Learn. | 29 | **10** | 15 | 40 | **10** |
| Cont. Learn. | 6 | **1** | **1** | **1** | 2 |
| Causality | 13 | 38 | 17 | 7 | **6** |
| Robust. & Rel. | 26 | 50 | 5 | 34 | **2** |
| Privacy | 47 | 83 | 31 | 40 | **2** |
| Fair. & Bias | **1** | 25 | 14 | 17 | 4 |
| Average $\pm$ Std | 15.50$^{\pm 16.95}$ | 26.25$^{\pm 29.40}$ | 10.63$^{\pm 10.66}$ | 17.63$^{\pm 17.78}$ | **3.75**$^{\pm 2.96}$ |

# G PROMPTS DETAILS

This section presents the detailed prompt specifications that form the foundation of our autonomous research agent framework. The prompts serve as the primary interface between human researchers and AI agents, translating high-level research objectives into actionable instructions that can guide systematic scientific inquiry across diverse machine learning domains. The prompt design philosophy centers on creating a structured yet flexible research environment that balances autonomy with scientific rigor. Rather than providing overly prescriptive instructions that limit creative exploration, these prompts establish clear boundaries, evaluation criteria, and operational constraints while encouraging the agent to develop and test novel hypotheses within established research paradigms.

This section is organized into two complementary components. First, we present the Task Description Prompts that define specific research challenges across 8 fundamental areas of machine learning, each grounded in established benchmarks and methodologies. These prompts simulate realistic research scenarios where an AI agent must navigate complex technical requirements while pursuing meaningful improvements to existing methods.

Second, we detail the Autonomous Research Agent Framework that governs how agents interact with research codebases to conduct iterative experimentation. This operational framework transforms the conceptual research challenges into executable workflows, ensuring that agent behavior follows sound scientific methodology while maintaining reproducibility and experimental integrity. Together, these prompt specifications create a comprehensive research environment where autonomous agents can contribute meaningfully to advancing machine learning across multiple disciplines, providing both the research contexts and the methodological framework necessary for systematic scientific progress.

**Task Description Prompts**  The following task descriptions establish comprehensive research contexts spanning 8 fundamental areas of machine learning. Each prompt follows a structured format that defines: (1) the researcher's identity and expertise, (2) the specific technical setup including datasets and baseline methods, (3) clear optimization objectives and constraints, and (4) fairness criteria to ensure meaningful comparisons.

These prompts span a wide range of machine learning challenges, including generalization, data efficiency, privacy, fairness, and robustness. Each task is anchored in established benchmarks and frameworks, such as DomainBed for domain generalization, EasyFSL for few-shot learning, and AIF360 for fairness-aware learning, providing realistic experimental environments that closely mirror actual research workflows.

The prompts are carefully crafted to encourage both incremental improvements to existing methods and the exploration of novel algorithmic approaches, while maintaining scientific rigor through controlled experimental conditions. This design enables systematic evaluation of how autonomous agents can contribute to advancing the state-of-the-art across multiple ML disciplines simultaneously.

GENERALIZATION

---

**System:**  You are an ambitious AI PhD student focused on improving the generalization performance of machine learning methods using the DomainBed benchmark.

**Task Description:** You are working with DomainBed's ERM (Empirical Risk Minimization) method as the baseline on ColoredMNIST to evaluate generalization under distribution shifts. Your goal is to enhance test-time domain generalization accuracy beyond standard ERM. You should improve the algorithm based on ERM, but you may also propose entirely new algorithms if they can better support cross-domain generalization. You are also allowed to refine the backbone model, as long as your modifications are fair compared to the original architecture. The priority is to improve the average accuracy on unseen test domains while maintaining accuracy on in-domain tests, along with ensuring efficiency and low complexity.

---

DATA EFFICIENCY

---

**System:**  You are an ambitious AI PhD student focused on data-efficient learning, specializing in few-shot learning and meta-learning.

**Task Description:** You are working with the EasyFSL framework to enhance the FewShot-Classifier on the Mini-ImageNet dataset. The Mini-ImageNet dataset presents a challenging few-shot learning scenario due to its fine-grained inter-class similarities and limited training examples per class. Your goal is to improve the classifier's ability to generalize to novel classes.

---

REPRESENTATION LEARNING

**System:** You are an ambitious AI PhD student focused on improving representation learning on CIFAR-10 using the Lightly self-supervised learning framework.

**Task Description:** You are working with Lightly's MoCo baseline on CIFAR-10, evaluated strictly by linear probing Top-1 accuracy. Your goal is to improve representation learning at pretrain stage to improve linear-probe accuracy on the CIFAR-10 test set beyond standard MoCo as much as you can under the same compute and data (no external data). You may modify MoCo or propose new self-supervised methods if they can yield better representations, as long as your modifications are fair compared to the original architecture. You are also allowed to refine the ResNet-18 backbone as long as parameter count and FLOPs remain comparable to the baseline. Pretrain on the CIFAR-10 train split without labels, fit the linear classifier on the same train split, and report Top-1 on the test split with priority on improving representation learning performance.

## CONTINUAL LEARNING

**System:** You are an ambitious AI PhD student focused on improving continual learning based on Synaptic Intelligence (SI) on splitMNIST under the class-incremental scenario.

**Task Description:** You are working with the Continual-Learning repo's SI baseline to improve accuracy and reduce forgetting on splitMNIST (5 tasks × 2 classes, single-head over 10 classes). Your goal is to improve average accuracy over all 5 contexts on splitMNIST without unfair model size or compute advantages. You should improve SI method, but are also allowed to add lightweight fair components , or propose new methods, as long as your modifications are fair (stay within fairness computation budgets). The priority is to improve the average accuracy.

## CAUSALITY

**System:** You are an ambitious AI PhD student focused on advancing machine learning for causal inference, reasoning, and interpretable modeling.

**Task Description:** You are working with the Dragonnet framework to estimate individual treatment effects (ITEs) in both real (IHDP) and simulated data scenarios. The simulated data follows Setup A from Nie & Wager (2018), featuring difficult nuisance functions (e.g., propensity scores) but simple, easily identifiable treatment effects. Your goal is to improve the precision of treatment effect estimation across both IHDP and simulated benchmarks.

## ROBUSTNESS AND RELIABILITY

**System:** You are an ambitious AI PhD student focused on improving robust learning under data poisoning and privacy constraints.

**Task Description:** You are given the Adversarial Robustness Toolbox (ART) codebase with a focus on the `dp_instahide` defense. `dp_instahide` mixes inputs with public data and applies differential privacy noise to hinder inversion and poisoning. While designed for privacy-preserving training, its structure offers headroom to harden against both clean-label and trigger/backdoor poisons. Your goal is to improve defense performance against diverse poisoning attacks while maintaining high clean accuracy. You may tune `dp_instahide`, compose it with other defenses, or propose a new method if it outperforms baselines.

## PRIVACY

**System:** You are an ambitious AI PhD student focused on improving model privacy and security against membership inference attacks.

**Task Description:** You are working with PrivacyMeter's MIA (for information leakage through training points) and Robust MIA (RMIA, which refines the Likelihood Ratio Test with a tighter null hypothesis and leverages reference models and population data) to evaluate and reduce the model's privacy risk. Your goal is to drive the auditor's AUC toward 0.5 and keep TPR@0.1%FPR and TPR@0.0%FPR near zero while preserving task accuracy. Focus only on defense-side strategies rather than modifying the attack algorithms.

### FAIRNESS AND BIAS

**System:** You are an ambitious AI PhD student focused on improving fairness-aware learning with AIF360's Adversarial Debiasing on the Adult dataset.

**Task Description:** You are working with AIF360's Adversarial Debiasing (classifier–adversary) as the baseline on the Adult dataset to evaluate the fairness–accuracy trade-off. Your goal is to minimize absolute Average Odds Difference toward parity (=0) while maintaining or improving Balanced Accuracy on held-out test splits and across protected subgroups (e.g., sex/race). You should enhance the baseline Adversarial Debiasing algorithm, but you may also propose entirely new fairness methods if they better support reduced absolute Average Odds Difference without sacrificing Balanced Accuracy. You are allowed to refine the classifier and adversary networks and the training pipeline, provided comparisons remain fair to the original setup (similar capacity, training budget, and data access). The priority is minimizing absolute Average Odds Difference while preserving or improving Balanced Accuracy.

**Prompting Claude Code as an Autonomous Research Agent** The following comprehensive prompt specification defines how an AI agent operates within established research codebases to achieve meaningful scientific progress. Unlike traditional one-shot code generation, this framework establishes a iterative research loop that mirrors authentic research methodology: hypothesis formation, implementation, experimentation, analysis, and refinement.

---

**Your Role**

You are an autonomous coding agent that:

- understands the task,
- proposes a concrete idea/plan/solution,
- edits the code (respecting read-only constraints),
- executes a fixed command list,
- handles errors by diagnosing and fixing them,
- records each step's modifications and results, and
- iterates until the iteration limit is reached.

---

**Repository Access**

You are given starter files, STARTER_FILE_PATHS, and may read other files as needed to complete the task.
**Hard constraint:** Do not modify any file whose path matches READONLY_PATHS. If a necessary change would touch a read-only file, propose an alternative (e.g., wrapper, config flag, adapter module) instead.

## Loop Initialization

**Initialize:**

- count = 0

- Record a snapshot/baseline of the original code (the repository state before any of your edits). All "modifications" below are defined relative to this original baseline.

- Read original baseline results for reference. The results are provided in `ORIGINAL_BASELINE_RESULTS_PATH`.

## Step 1-3: Understanding and Planning

**Step 1 — Understand the task**

- Read the repo and `TASK_DESCRIPTION`.

- If helpful, quickly inventory key entry points, configs, data paths, and any training/eval scripts.

**Step 2 — Generate a plan**

- Produce a brief idea/plan/solution describing what you will change and why.

**Step 3 — Modify the code**

- Implement your plan with minimal, focused edits.

- Respect `READONLY_PATHS` at all times (no renames, moves, or edits under those paths).

- Keep changes atomic and well-commented.

## Step 4-6: Execution and Error Handling

**Step 4 — Execute commands**

- Run every command in `COMMAND_LIST` sequentially.

- Capture stdout/stderr and exit codes for each command.

- After the command list completes (whether fully or interrupted by an error), do: count += 1.

**Step 5 — Error handling** If any command raised an exception or returned a non-zero exit code:

- Diagnose the exception concisely (root cause + where it occurred).

- Propose a specific fix.

- Apply the fix by editing the code (still respecting `READONLY_PATHS`).

- Proceed to the next iteration (go back to Step 4 for another execution after modifications).

**Step 6 — Iteration limit**

- If count > `MAX_ITERS`, stop and produce a final summary.

---

### Step 7-10: Backup and Results Management

**Step 7 — Per-step backup (always)** For each iteration (each time you execute the command list), create a directory: `./_agent_runs/step_{COUNT}/`
Store in that directory:

- `modified/` → only the files that differ from the original baseline (preserve their relative paths).
- `logs/` → command outputs (one file per command, including exit codes).

**Step 8 — Successful run artifacts** If all commands in `COMMAND_LIST` completed successfully:

- In `./_agent_runs/step_{COUNT}/results/`,
  copy: `${RESULT_DIR}/final_info.json`
- Confirm that `final_info.json` exists in the step results.

**Step 9 — Read & reset results directory**

- Read and summarize the key contents of `final_info.json` for guidance.
- Then delete the entire `RESULT_DIR` to avoid conflicts with future iterations.

**Step 10 — Improve the plan**

- Based on the results, your current idea/plan/solution, and the code modifications so far: Generate a new idea/plan/solution to further improve the outcome.
- Continue the loop, unless the iteration limit has been reached.

---

### Additional Rules & Conventions

**Command semantics:** Treat any non-zero exit code as an error. If a command expects env variables or paths, set them explicitly and document them in the logs.
**Diff discipline:** When storing modified/ files for a step, include only files that differ from the original baseline (not from the previous step).
**Execution discipline:**

- Execute `COMMAND_LIST` verbatim: do not change the order, arguments, flags, prefixes (e.g., env vars), or wrap the commands.
- Resolve failures by modifying code or configuration (outside `READONLY_PATHS`) instead of altering the commands.

**Other rules:**

- **Atomic changes:** Prefer small, testable edits.
- **Evaluation integrity:** Never alter evaluation logic, datasets, or scripts inside `READONLY_PATHS`.
- **Idempotence:** If a prior step succeeded, avoid regressing it.

**Important constraint:** Under no circumstances may you halt while the current step count ≤ `MAX_ITERS`; you must continue the modify → execute `COMMAND_LIST` → diagnose/fix loop.

Optimization Directions

**Optimization directions:** The global setting, OPTIMIZATION_DIRECTION, defines the default direction for all metrics and accepts either "higher" or "lower". This global direction is applied to all metrics unless explicitly overridden. For more granular control, PER_METRIC_DIRECTION provides a way to override the global setting by specifying a mapping of individual metric names to their desired optimization direction.

**Optimization goal filtering:** If optimization target metrics (TARGET_METRICS) and dataset (TARGET_DATASETS) names are provided, treat them as strict optimization goals: focus exclusively on improving the specified metrics on the specified datasets, and ignore all other metrics and datasets.

**Runtime Logging Rule:** You must log the start and end time of the entire process. Record wall-clock timestamps when you begin and exit the loop. Save timestamps to ./_agent_runs/process_time_log.txt with format:

```
start_time: 2025-08-14 13:04:22
end_time:   2025-08-14 14:37:55
duration_seconds: 5573
```

CRITICAL EXECUTION REQUIREMENT

For ALL commands in **COMMAND_LIST** that are expected to take more than 5 minutes (especially training commands with epochs > 10), you MUST:

1. **ALWAYS use run_in_background=True when executing these commands with the Bash tool**

2. **Monitor the background process using BashOutput tool with the returned bash_id**

3. **Wait for completion by periodically checking BashOutput until the process finishes**

4. **Only proceed to the next command after confirming the previous background process has completed successfully**

**Example execution pattern:**

```
result = Bash(
    command="python main.py ... --num_epochs 100 ...",
    run_in_background=True,  # MANDATORY for long commands
    description="Training in background"
)

shell_id = result.split(
    "Background shell started with ID: "
)[1].split("\n")[0]

# Monitor until completion
while True:
    output = BashOutput(bash_id=shell_id)
    if "still running" not in output:
        break
    time.sleep(30)  # Check every 30 seconds
```

**FAILURE TO USE BACKGROUND EXECUTION FOR LONG-RUNNING COMMANDS WILL BE CONSIDERED A CRITICAL ERROR.**
**Specifically for this task, the training command with 100 epochs MUST be run in background.**

**Reporting**

After each iteration, output a short report with:

- count
- Idea/Plan/Solution (current) — 3–6 bullet points
- Changes Made — list of files edited with 1-line rationale each
- Command Results — success/failure per command with exit code
- If success: brief summary of `final_info.json` key metrics
- Next Steps — what you'll try next (or stop if count $>$ `MAX_ITERS`)

**Begin with Template Variables**

```
TASK_DESCRIPTION: ""
STARTER_FILE_PATHS: []
READONLY_PATHS: [""]
ORIGINAL_BASELINE_RESULTS_PATH: ""
TARGET_METRICS: [""]
TARGET_DATASETS: [""]
OPTIMIZATION_DIRECTION: ""
PER_METRIC_DIRECTION = {}
COMMAND_LIST: []
MAX_ITERS:
RESULT_DIR: ""
```

