# OpenReview forum: "A Benchmark for Automatic ML Research Agents Highlighting the Importance of Exploration Breadth"
_ICLR.cc/2026/Conference — Submitted to ICLR 2026_

### Official Review · Reviewer_NpRn · 2025-10-26

**Soundness:** 2
**Presentation:** 2
**Contribution:** 2
**Rating:** 4
**Confidence:** 3

**Summary:**

This paper introduces FML-bench, a benchmark designed to evaluate automatic machine learning research agents on fundamental ML problems rather than engineering-style tasks.
The benchmark includes eight core tasks (e.g., generalization, continual learning, causality, fairness) and evaluates agents across five dimensions: utility, diversity, academic contribution rate, cost, and step success rate.
Experiments on three representative agent frameworks demonstrate that agents employing broader exploration strategies tend to outperform those focusing on narrow, deep refinement.

**Strengths:**

• The authors correctly observe that existing benchmarks for ML agents primarily assess engineering execution rather than scientific research competence, which is an insightful and valuable perspective.

• Building on this observation, the paper introduces several thoughtful design choices — for instance, providing baseline repositories so that agents can start from existing codebases rather than building projects from scratch, thereby reducing dependence on pure coding ability.

• The authors make a clear effort to ensure benchmark usability and accessibility. I checked the provided code link, and it reflects a well-organized release and attention to potential community impact.

**Weaknesses:**

• As reflected in the title, the main contribution and conclusion “The Importance of Exploration Breadth” may be heavily conditioned on the specific experimental settings. The entire setup is based on combinational innovation of the form A + B = C, which naturally favors “broad exploration” strategies. This could mislead future research directions, encouraging agent systems to maximize benchmark scores by generating superficial combinations instead of pursuing deeper innovation.

• Equation (1) seems to be a conceptual pseudo-formula rather than an operational objective. The five metrics have inconsistent scales, and summing them directly would let high-magnitude terms dominate. The authors do not explain how coefficients such as lambda or beta are chosen, leaving this formulation vague.

• Some metric designs appear rather superficial and arbitrary. For example, for Utility, why did the authors not normalize improvements by percentage? Different tasks have different baseline performance levels. A 20% increase on an unsaturated dataset can be much easier than a 1% gain on ImageNet, even though the latter is more meaningful (I understand this does not change the experiment’s internal conclusion). Diversity relies only on semantic similarity, which may be unstable in large samples. If two pieces of code have identical logic but only differ in variable or function names, would their similarity be high or low? The notion of Academic Contribution Rate raises fundamental questions of subjectivity: How is “academic contribution” actually quantified? For example, ReLU is, in essence, just a simple piecewise linear function. Should that be considered an academic innovation or an engineering refinement? And what about data augmentation? Is it categorized as an engineering technique or as a scientific contribution?

• In Section 5.3.2, beyond correlation coefficients, significance tests should also be conducted, especially given the limited sample size (it is unclear how many samples were actually used for correlation calculation).

• The benchmark examples may not be very discriminative. A benchmark should separate strong methods from weak ones, yet in Tables 2 and 3, the scores are either all high or all low, with little variance across systems. This diminishes its usefulness as a benchmark. Moreover, considering the fast pace of model updates from companies like OpenAI, the benchmark risks becoming obsolete quickly.

**Questions:**

• The authors may consider removing the Academic Contribution Rate metric, which is potentially problematic, especially given that it is derived from LLM-based judgments. Introducing such an automated measure of “scientific contribution” could create conceptual and ethical risks for the community.

• Since the benchmark is described as easily extensible, the authors could consider scaling it up and selecting tasks more carefully so that different systems fall within a reasonable range, rather than the current situation where all perform either very well or very poorly.

---

> ### Author Response · Authors · 2025-12-02
> **Response to Reviewer  NpRn [1/4]**
>
> We thank the reviewer for the constructive feedback and address each point below:
>
> > **Q.1.:** “As reflected in the title, the main contribution and conclusion “The Importance of Exploration Breadth” may be heavily conditioned on the specific experimental settings. The entire setup is based on combinational innovation of the form A + B = C, which naturally favors “broad exploration” strategies. This could mislead future research directions, encouraging agent systems to maximize benchmark scores by generating superficial combinations instead of pursuing deeper innovation.”
> >
>
> We thank the reviewer for raising this concern. We would like to clarify that our experimental setting is intended to incentivize agents to propose approaches that enhance task performance (i.e., utility). **Any effective proposal by agent that contributes to such improvement is encouraged**, including the development and refinement of new methods, rather than solely combinational innovations of the form *A + B = C*. Under this setting, the TheAiScientist, which adopts a strategy emphasizing exploration breadth, achieves relatively superior performance compared to other agents, suggesting the importance of exploration breadth.
>
> > **Q.2.:** “Equation (1) seems to be a conceptual pseudo-formula rather than an operational objective. The five metrics have inconsistent scales, and summing them directly would let high-magnitude terms dominate. The authors do not explain how coefficients such as lambda or beta are chosen, leaving this formulation vague.”
> >
>
> We thank the reviewer for pointing out this issue. We will revise this section to present these as five independent metrics rather than a combined formulation. Specifically, we will clarify that Utility is the sole metric reflecting the final performance objective, while the other four metrics serve as descriptive and analytical indicators.
>
> > **Q.3.1.:** “For example, for Utility, why did the authors not normalize improvements by percentage? Different tasks have different baseline performance levels. A 20% increase on an unsaturated dataset can be much easier than a 1% gain on ImageNet, even though the latter is more meaningful (I understand this does not change the experiment’s internal conclusion). “
> >
>
> We appreciate the reviewer's suggestion regarding the intuitive representation of agent performance. However, we chose not to normalize improvements across different tasks for two primary reasons:
>
> - First, different tasks employ heterogeneous evaluation metrics (e.g., recall versus MSE), making direct normalization technically challenging.
> - Second, we lack a principled method to quantify the relative difficulty of datasets across tasks, which would be necessary for difficulty adjusted normalization.
>
> Consequently, we report raw metric values directly and compare agent performance by the number of best results each agent achieves.
>
> > **Q.3.2.:** “Diversity relies only on semantic similarity, which may be unstable in large samples. If two pieces of code have identical logic but only differ in variable or function names, would their similarity be high or low? “
> >
>
> We employ GraphCodeBERT (line#419-423, #773-776), which **leverages data-flow graphs and code structure rather than surface tokens alone**. Consequently, code snippets with identical logic but different variable/function names receive high similarity scores due to their shared structural patterns. Such pairs will be treated as low diversity in our Diversity metric. We will add clarification on this point in the main text.

---

> ### Author Response · Authors · 2025-12-02
> **Response to Reviewer  NpRn [2/4]**
>
> > **Q.3.3.:** “The notion of Academic Contribution Rate raises fundamental questions of subjectivity: How is “academic contribution” actually quantified? For example, ReLU is, in essence, just a simple piecewise linear function. Should that be considered an academic innovation or an engineering refinement? And what about data augmentation? Is it categorized as an engineering technique or as a scientific contribution?”
> >
>
> To clarify, the **academic contribution rate metric compares each step with the baseline method rather than comparing consecutive steps with each other** . This metric is designed to encourage agents to propose new ML methods that differ from the baseline rather than merely tuning hyperparameters (engineering modification). Any method that introduces a new machine learning idea (e.g. a novel objective or inductive bias) is classified as an academic modification, regardless of its complexity (lines#454-464).
>
> For example, if step_k proposes a new method and is classified as an academic step, then step_k+1, which refines step_k, is also classified as an academic step because both are compared against the baseline rather than against each other.
>
> Therefore, simply adding ReLU to the baseline or modifying network depth constitutes an engineering modification. Similarly, if step_k introduces a novel architecture and step_k+1 adjusts it by adding ReLU, both steps qualify as academic modifications.
>
> We will revise the manuscript to provide a clearer description of this metric and its interpretation.
>
> > **Q.4.:** “In Section 5.3.2, beyond correlation coefficients, significance tests should also be conducted, especially given the limited sample size (it is unclear how many samples were actually used for correlation calculation).”
> >
>
> We thank the reviewer for this suggestion. We have increased the sample size from 5 (original) to 15 pairs (3 rounds × 5 agent configurations) to enhance statistical robustness for each task’s correlation computation. On the revised FML-bench with 3 hard tasks replaced (addressing Q.5), we conducted new correlation analysis between exploration diversity and outcome quality.
>
> Our findings show that, across five tasks, we observed consistent positive correlations between exploration diversity and performance, with p-values indicating significant, borderline significant, or suggestive evidence of this relationship. It indicates that, for the five tasks, higher exploration diversity was associated with better outcomes.
>
> | Task | p-value | Correlation | Significance |
> | --- | --- | --- | --- |
> | Generalization | 0.113738 | **0.4418** | suggestive evidence |
> | Data Eff. | 0.012058 | **0.6287** | significant |
> | Rep. Learn. | 0.135768 | **0.4036** | suggestive evidence |
> | Cont. Learn. | 0.000186 | **0.8374** | significant |
> | Causality | 0.466795 | -0.2036 | not significant |
> | Robust. & Rel. | 0.194413 | -0.3846 | not significant |
> | Privacy | 0.647916 | -0.1286 | not significant |
> | Fair. & Bias | 0.051652 | **0.5292** | borderline significant |
> - p < 0.05 (significant); p < 0.10 (borderline significant); p < 0.15 (suggestive evidence); p >= 0.15 (not significant)
> - We negated scores for tasks where lower values indicate better performance, ensuring that positive correlations consistently represent improved performance with increased diversity.
>
> We will update the main text accordingly.

---

> ### Author Response · Authors · 2025-12-02
> **Response to Reviewer  NpRn [3/4]**
>
> > **Q.5.:** “The benchmark examples may not be very discriminative. A benchmark should separate strong methods from weak ones, yet in Tables 2 and 3, the scores are either all high or all low, with little variance across systems. This diminishes its usefulness as a benchmark. Moreover, considering the fast pace of model updates from companies like OpenAI, the benchmark risks becoming obsolete quickly.”
> >
>
> We thank the reviewer for identifying the insufficient difficulty in certain tasks. We have increased the difficulty of three tasks (Fairness & Bias, Causality, Robustness & Reliability) to better differentiate agent performance and provide headroom for future improvements. Specifically:
>
> - **Fairness & Bias** (repo: AIF360): We replaced the Adult dataset with the more challenging COMPAS dataset [1] and imposed an accuracy threshold (set at baseline performance) to exclude solutions that sacrifice overall accuracy for fairness gains.
> - **Causality** (repo: CausalML): We replaced the synthetic dataset (constructed from [2]) with the more difficult IHDP dataset [3] and changed the metric from absolute percentage error of ATE to mean absolute error for clearer differentiation.
> - **Robustness & Reliability** (repo: ART): We introduced additional backdoor attack methods (edge-based triggers and various distributed attack patterns) to increase defense difficulty.
>
> The results on the updated FML-bench are as follows:
>
> | **ML Problems** | **Baseline** | **TheAIScientist - GPT-5** | **TheAIScientist - G2.5-Pro** | **AIDE - GPT-5** | **AIDE - G2.5-Pro** | **Claude Code - Opus-4.1** |
> | --- | --- | --- | --- | --- | --- | --- |
> | Generalization (↑) | 0.2254 | **0.5036** | 0.3252 | 0.2254 | 0.2254 | **0.5036** |
> | Data Eff. (↑) | 0.6547 | 0.7689 | **0.8231** | 0.6547 | 0.6547 | 0.6571 |
> | Rep. Learn. (↑) | 0.7562 | 0.7796 | **0.8597** | 0.8469 | 0.8466 | 0.7725 |
> | Cont. Learn. (↑) | 0.271 | 0.4281 | **0.7810** | 0.4369 | 0.3660 | 0.2337 |
> | Causality (↓) | 1.1445 | 1.0063 | 0.9925 | 0.9683 | **0.9549** | 0.9840 |
> | Robust. & Rel. (↑) | 0.4848 | 0.9311 | 0.9205 | 0.9174 | **0.9633** | 0.8921 |
> | Privacy (↓) | 0.8114 | 0.4908 | **0.1750** | 0.4882 | 0.4814 | 0.4892 |
> | Fair. & Bias (↓) | 0.3787 | 0.0603 | 0.1002 | **0.0385** | 0.0917 | 0.3787 |
>
> The three modified tasks now better differentiate agent performance while maintaining consistent performance trends: TheAIScientist outperforms AIDE, which outperforms Claude Code.
>
> We will update the main text accordingly.
>
> [1] J. Angwin, J. Larson, S. Mattu, L. Kirchner, “Machine bias: There’s software used across the country to predict future criminals. And it’s biased against blacks,” ProPublica, 23 May 2016
>
> [2] Nie, Xinkun, and Stefan Wager. "Quasi-oracle estimation of heterogeneous treatment effects." *Biometrika* 108.2 (2021): 299-319.
>
> [3] Hill, Jennifer L. "Bayesian nonparametric modeling for causal inference." *Journal of Computational and Graphical Statistics* 20.1 (2011): 217-240.

---

> ### Author Response · Authors · 2025-12-02
> **Response to Reviewer  NpRn [4/4]**
>
> > **Q.6.:** “The authors may consider removing the Academic Contribution Rate metric, which is potentially problematic, especially given that it is derived from LLM-based judgments. Introducing such an automated measure of “scientific contribution” could create conceptual and ethical risks for the community.”
> >
>
> We thank the reviewer for this suggestion. As explained in Q.3.3, academic contribution rate compares each step against the baseline method rather than the previous step, intentionally encouraging agents to propose methods beyond simple hyperparameter tuning. Any new approach, whether simple or complex, qualifies as an academic modification. We will remove academic contribution rate from the objective function and instead **present it as an independent analytical metric**.
>
> Moreover, to address concerns regarding LLM-based judgments,  we have conducted a **human evaluation** to validate the accuracy of these assessments. Three human annotators each evaluated 200 samples (5 agents × 40 samples per agent). The results are presented in the following table:
>
> | Model | Accuracy of LLM Evaluation (Human-evaluated) | LLM-Judged Academic Contribution Ratio | Human-Judged Academic Contribution Ratio |
> | --- | --- | --- | --- |
> | TheAIScientist - GPT-5 | 0.7982 ± 0.0992 | 0.83 ±  0.27 | 0.8947 ± 0.1489 |
> | TheAIScientist - Gemini-2.5-pro | 0.7807 ± 0.0969 | 0.78 ±  0.24 | 0.7281 ± 0.1183 |
> | AIDE - GPT-5 | 0.8198 ± 0.1215 | 0.84 ±  0.20 | 0.7928 ± 0.1770 |
> | AIDE - Gemini-2.5-pro | 0.8120 ± 0.0604 | 0.65 ±  0.34 | 0.6239 ± 0.0944 |
> | ClaudeCode - Opus-4.1 | 0.7701 ± 0.1445 | 0.25  ± 0.36 | 0.5057 ± 0.1445 |
> | Overall mean | 0.7962 ± 0.0186 | 0.67 ± 0.2206 | 0.7091 ± 0.1345 |
> - Our human evaluation reveals that the LLM with Qwen3-235b-a22b achieves an overall accuracy of 79.62% in its judgments.
> - Additionally, we report the human-evaluated academic contribution ratios in the table above, providing a more reliable assessment of agent performance.
>
> We will update the main text accordingly.
>
> > **Q.7.:** “Since the benchmark is described as easily extensible, the authors could consider scaling it up and selecting tasks more carefully so that different systems fall within a reasonable range, rather than the current situation where all perform either very well or very poorly.”
> >
>
> We thank the reviewer for this suggestion. As addressed in Q.5, we have increased the difficulty of three tasks. Different agents now demonstrate better performance differentiation across all eight tasks.

---

### Official Review · Reviewer_mYE9 · 2025-11-01

**Soundness:** 3
**Presentation:** 3
**Contribution:** 3
**Rating:** 6
**Confidence:** 4

**Summary:**

The paper introduces the FML-benchmark, which can evaluate the automatic machine learning research agents on eight diverse and fundamental machine learning research problems. The proposed benchmark is more general and can extend to real-world GitHub repositories. The paper also provides a unified evaluation framework with five metrics. The paper also evaluates the state-of-the-art automatic research agents on FML-bench,

**Strengths:**

1. Instead of focusing on the engineering perspective, this paper focuses on fundamental ML problems grounded on real-world codebases, including eight categories. All evaluated agents are SOTA.
2. The paper provides a unified evaluation framework to help formalize the problem.  The paper provides a five-dimensional evaluation protocol covering utility, diversity, academic contribution rate, cost, and step success rate. Apart from pure implementation perspective, the paper also focuses on the diversity, academic contribution, and time. The paper also provides standard deviation for each metric. Those metrics include both quantitative (formula-based) and qualitative metrics (LLM-based).
3. The paper provides benchmark code, experiment prompts, and configuration files on an anonymous GitHub repo. The visualization of this paper is very good and intuitive.

**Weaknesses:**

1. Some details are missing. For the 8 diverse tasks, it might be better to provide detailed statistics for each of the tasks, such as the number of repositories and task settings. Most of those details are put in the appendix. It would be better to move them (e.g., task definition, metrics definition) into the main text.
2. The evaluation integrity part is not very rigorous. Some automatic comparison between the evaluation files might be needed. Including a comparison with human evaluation on those corresponding evaluation metrics can further strengthen the paper.
3. The paper did not contain an error analysis section, which will be very valuable for readers to further improve the existing AI agent frameworks.

**Questions:**

See weakness

---

> ### Author Response · Authors · 2025-12-02
> **Response to Reviewer  mYE9 [1/2]**
>
> We thank the reviewer for the constructive feedback and address each point below:
>
> > **Q.1.:** “Some details are missing. For the 8 diverse tasks, it might be better to provide detailed statistics for each of the tasks, such as the number of repositories and task settings. Most of those details are put in the appendix. It would be better to move them (e.g., task definition, metrics definition) into the main text.”
> >
>
> We thank the reviewer for this suggestion. To clarify, each of the eight tasks in FML-bench utilizes a single repository. We will enhance the presentation of this section by relocating the detailed task settings and metric definitions from the appendix to the main text.
>
> > **Q.2.:** “The evaluation integrity part is not very rigorous. Some automatic comparison between the evaluation files might be needed. Including a comparison with human evaluation on those corresponding evaluation metrics can further strengthen the paper.”
> >
>
> We thank the reviewer for highlighting this important concern. To ensure evaluation integrity, our benchmark **protects evaluation files** as read-only and protected from agent modification (line#292, #301-303). Agents must utilize these protected evaluation files to assess their proposed methods, thereby guaranteeing the integrity of the evaluation process.
>
> Regarding the use of LLM-based judgment for evaluating academic contribution ratios, we have conducted a human evaluation to validate the accuracy of these assessments. Three human annotators each evaluated 200 samples (5 agents × 40 samples per agent). The results are presented in the following table:
>
> | Model | Accuracy of LLM Evaluation (Human-evaluated) | LLM-Judged Academic Contribution Ratio | Human-Judged Academic Contribution Ratio |
> | --- | --- | --- | --- |
> | TheAIScientist - GPT-5 | 0.7982 ± 0.0992 | 0.83 ±  0.27 | 0.8947 ± 0.1489 |
> | TheAIScientist - Gemini-2.5-pro | 0.7807 ± 0.0969 | 0.78 ±  0.24 | 0.7281 ± 0.1183 |
> | AIDE - GPT-5 | 0.8198 ± 0.1215 | 0.84 ±  0.20 | 0.7928 ± 0.1770 |
> | AIDE - Gemini-2.5-pro | 0.8120 ± 0.0604 | 0.65 ±  0.34 | 0.6239 ± 0.0944 |
> | ClaudeCode - Opus-4.1 | 0.7701 ± 0.1445 | 0.25  ± 0.36 | 0.5057 ± 0.1445 |
> | Overall mean | 0.7962 ± 0.0186 | 0.67 ± 0.2206 | 0.7091 ± 0.1345 |
> - Our human evaluation reveals that the LLM with Qwen3-235b-a22b achieves an overall accuracy of 79.62% in its judgments.
> - Additionally, we report the human-evaluated academic contribution ratios in the table above, providing a more reliable assessment of agent performance.
>
> We will incorporate these human evaluation results into the manuscript accordingly.

---

> ### Author Response · Authors · 2025-12-02
> **Response to Reviewer  mYE9 [2/2]**
>
> > **Q.3.:** “The paper did not contain an error analysis section, which will be very valuable for readers to further improve the existing AI agent frameworks.”
> >
>
> We thank the reviewer for this valuable suggestion. We conducted **worst-case** and **best-case analyses** examining modification distribution patterns to provide deeper insights into agent and LLM performance characteristics on FML-bench. Worst cases represent steps with the poorest performance in each experimental round, while best cases represent steps with the highest performance.
>
> **Worst Case Analysis - Modification Distribution by Agents:**
>
> - **The AI Scientist**: Strongly prefers algorithmic modifications (56.2%), with no implementation errors, indicating robust execution.
> - **AIDE**: Shows balanced distribution but has higher implementation and execution issues (31.2%).
> - **Claude Code**: Exhibits high parameter configuration rate (37.5%), matching its algorithmic modification rate.
>
>
>     | Modification Type | The AI Scientist | AIDE | Claude Code |
>     | --- | --- | --- | --- |
>     | Algorithmic Modification | 56.2% (9) | 37.5% (6) | 37.5% (3) |
>     | Structural Modification | 31.2% (5) | 25.0% (4) | 12.5% (1) |
>     | Parameter Configuration | 12.5% (2) | 6.2% (1) | 37.5% (3) |
>     | Implementation & Execution | 0.0% (0) | 31.2% (5) | 12.5% (1) |
>
> **Worst Case Analysis - Modification Distribution by LLMs:**
>
> - **Gemini-2.5-pro**: Extremely strong algorithmic modification tendency (68.8%), preferring core algorithm changes.
> - **GPT-5**: Primarily focuses on structural modifications (50.0%), favoring architecture adjustments.
> - **Opus-4.1**: Maintains balanced distribution between parameter configuration and algorithmic modifications.
>
>
>     | Modification Type | GPT-5 | Gemini-2.5-pro | Opus-4.1 |
>     | --- | --- | --- | --- |
>     | Algorithmic Modification | 25.0% (4) | 68.8% (11) | 37.5% (3) |
>     | Structural Modification | 50.0% (8) | 6.2% (1) | 12.5% (1) |
>     | Parameter Configuration | 6.2% (1) | 12.5% (2) | 37.5% (3) |
>     | Implementation & Execution | 18.8% (3) | 12.5% (2) | 12.5% (1) |
>
> **Best Case Analysis - Modification Distribution by Agents:**
>
> - **The AI Scientist**: Strong algorithmic innovation tendency, with vast majority targeting core algorithmic logic.
> - **AIDE**: Strong algorithmic innovation with some hyperparameter tuning.
> - **Claude Code**: Unique preference for architectural modifications (e.g., adding layers, changing normalization).
>
>
>     | Modification Type | The AI Scientist | AIDE | Claude Code |
>     | --- | --- | --- | --- |
>     | Algorithmic Modification | 75.0% (12) | 83.3% (10) | 37.5% (3) |
>     | Structural Modification | 12.5% (2) | 0.0% (0) | 37.5% (3) |
>     | Data Processing | 12.5% (2) | 16.7% (2) | 0.0% (0) |
>     | Parameter Configuration | 0.0% (0) | 0.0% (0) | 25.0% (2) |
>
> **Best Case Analysis - Modification Distribution by LLMs:**
>
> - **GPT-5**: Overwhelmingly focuses on algorithmic modifications (100.0%).
> - **Gemini-2.5-pro**: only model with significant data augmentation attempts (28.6%).
> - **Opus 4.1**: Balanced attention to both architectural and algorithmic modifications.
>
>
>     | Modification Type | GPT-5 | Gemini-2.5-pro | Opus-4.1 |
>     | --- | --- | --- | --- |
>     | Algorithmic Modification | 14 (100.0%) | 8 (57.1%) | 3 (37.5%) |
>     | Structural Modification | 0 (0.0%) | 2 (14.3%) | 3 (37.5%) |
>     | Data Processing | 0 (0.0%) | 4 (28.6%) | 0 (0.0%) |
>     | Parameter Configuration | 0 (0.0%) | 0 (0.0%) | 2 (25.0%) |
>
> We will incorporate these analyses into the manuscript to provide insights for improving AI agent frameworks.

---

### Official Review · Reviewer_9hah · 2025-11-03

**Soundness:** 3
**Presentation:** 3
**Contribution:** 3
**Rating:** 4
**Confidence:** 4

**Summary:**

This paper introduces FML-bench, a new benchmark for ML engineering. The motivation is to provide a diverse set of real-world codebases and focus the evaluation on the ML rather than writing code from scratch or non-ML skills. It is also noted that the ability of an agent to generate diverse hypotheses can be correlated with utility.

**Strengths:**

I appreciate that the paper is thoughtful about a more careful curation of ML tasks which both covers different aspects of ML as well as focuses more on the ML side rather than general software engineering.
Looking at diversity and academic contribution rate is interesting and provides a fuller picture of what the agents are doing beyond accuracy.
The evaluations include a reasonable set of modern models and scaffolds.

**Weaknesses:**

In section 3, it would be good to separate the evaluation metrics into ones that are objective (utility and cost) and the ones that I take to be more like shaping rewards (e.g., diversity), which don't matter in themselves, but might be helpful for achieving the true objectives.

Section 5.3.3: It's unclear why one wants to encourage academic contribution rates. If you want to improve an ML approach, sometimes it might be better to stick with a simpler method and do better tuning as opposed to aim for a fancier method (an "academic contribution").

It seems like there are two contributions in this paper: (i) introducing the new benchmark (FML-bench) and (ii) studying the diversity and academic contributions of various methods. For example, one could study (ii) with existing benchmarks besides FML-bench.

While I appreciate the thoughtfulness of thinking about the composition of FML-bench, the 8 categories are fairly broad, and I worry that having a single task to represent a big topic like "fairness & bias" conflates the general category with the specifics of the problem setup, leading to potentially misleading conclusions (e.g., causality is easier than privacy).

**Questions:**

Questions
 - Was there any prompt optimization and hyperparameter tuning done?  If so, what was the basis for this?
 - Regarding the 8 tasks, I'm looking at Appendix B (I think the citations of the tasks/dtaasets could appear in the main body), but I still have some questions on how the tasks were defined? how was the difficulty determined? How do you ensure lack of train-test contamination? This is important since most of these datasets are old and likely show up in the training dataset.

---

> ### Author Response · Authors · 2025-12-02
> **Response to Reviewer  9hah [1/2]**
>
> We thank the reviewer for the constructive feedback and address each point below:
>
> > **Q.1.:** “In section 3, it would be good to separate the evaluation metrics into ones that are objective (utility and cost) and the ones that I take to be more like shaping rewards (e.g., diversity), which don't matter in themselves, but might be helpful for achieving the true objectives.”
> >
>
> We thank the reviewer for this insightful suggestion. We will position utility as the primary objective metric for evaluating agent performance. Other metrics like diversity and academic contribution rate will be reclassified as analytical metrics for characterizing agent exploration behavior (also could potentially serve as a shaping reward), rather than being treated as objectives.
>
> > **Q.2.:** “Section 5.3.3: It's unclear why one wants to encourage academic contribution rates. If you want to improve an ML approach, sometimes it might be better to stick with a simpler method and do better tuning as opposed to aim for a fancier method (an "academic contribution").”
> >
>
> We thank the reviewer for identifying this lack of clarity in our description.
>
> To clarify, the **academic contribution rate metric compares each step with the baseline method rather than comparing consecutive steps with each other**. This metric is designed to encourage agents to propose new ML methods that differ from the baseline rather than merely tuning hyperparameters (engineering modification). Any method that introduces a new machine learning idea (e.g. a novel objective or inductive bias) is classified as an academic modification, regardless of its complexity.
>
> For example, if step_k proposes a new method and is classified as an academic step, then **step_k+1, which refines step_k, is also classified as an academic step** because both are compared against the baseline rather than against each other.
>
> We will revise the manuscript to provide a clearer description of this metric and its interpretation.
>
> > **Q.3.:** "It seems like there are two contributions in this paper: (i) introducing the new benchmark (FML-bench) and (ii) studying the diversity and academic contributions of various methods. For example, one could study (ii) with existing benchmarks besides FML-bench."
> >
>
> We thank the reviewer for recognizing our contributions. However, we would like to clarify a potential misunderstanding regarding the two metrics of agent exploration diversity and academic contribution rate. **Our contribution of the two metrics includes proposing these two  metrics and investigating their correlation with agent performance.** This extends beyond merely studying these metrics (mentioned in the question).
>
> Furthermore, **existing research agent benchmarks** to our knowledge (including MLE-bench, MLAgentBench, DSBench, and ML-Dev-Bench) primarily focus their evaluation on final task performance metrics (e.g., accuracy, recall) and computational cost, while **overlooking the characteristics of agents' internal iterative processes**.
>
> **In contrast, our benchmark introduces the diversity and academic contribution rate metrics specifically to analyze agent behavior** throughout the entire research iteration process, providing insights into how agents explore and develop solutions rather than solely evaluating their final outcomes.
>
> We will revise the manuscript accordingly to better clarify this.
>
> > **Q.4.:** “While I appreciate the thoughtfulness of thinking about the composition of FML-bench, the 8 categories are fairly broad, and I worry that having a single task to represent a big topic like "fairness & bias" conflates the general category with the specifics of the problem setup, leading to potentially misleading conclusions (e.g., causality is easier than privacy).”
> >
>
> We thank the reviewer for this important observation. To clarify, FML-bench comprises eight diverse tasks that span eight different research domains, rather than attempting to comprehensively evaluate agent performance within each domain. Each task is intended to represent a distinct research challenge from its respective field, and we do not claim that individual tasks capture the full breadth or complexity of their corresponding domains. We will revise the manuscript to better articulate this and avoid potential misinterpretation or misleading conclusions.
>
> > **Q.5.:** “Was there any prompt optimization and hyperparameter tuning done? If so, what was the basis for this?”
> >
>
> We did not perform any manual prompt optimization or hyperparameter tuning. As a benchmark platform, we only provide task descriptions during agent initialization. The decision to employ prompt optimization or hyperparameter tuning techniques is left entirely to the individual agents' design and their internal reasoning and iteration processes.

---

> ### Author Response · Authors · 2025-12-02
> **Response to Reviewer  9hah [2/2]**
>
> > **Q.6.1.:** “I think the citations of the tasks/datasets could appear in the main body”
> >
>
> We thank the reviewer for this suggestion. We will incorporate the detailed task and dataset descriptions from the Appendix into Section 5.1 (benchmark task setting description) of the main text.
>
> > **Q.6.2.:** “how the tasks were defined? how was the difficulty determined?”
> >
>
> Our task definition process proceeded as follows:
>
> - We identified eight representative research problems spanning diverse domains in machine learning.
> - For each domain, we selected widely-used benchmark datasets that are both representative of the field and computationally tractable, with the constraint that baseline methods can complete training and evaluation within 2 hours.
> - Among candidates meeting these criteria, we prioritized datasets that present greater research challenges to better differentiate agent capabilities.
> - Finally, we implemented simple yet effective baseline methods for each task. These baselines serve as starting points for agents, providing sufficient room for improvement while enabling meaningful comparisons of different agents' research capabilities.
>
> We will incorporate this process description into the manuscript.
>
> > **Q.6.3.:**  “How do you ensure lack of train-test contamination? This is important since most of these datasets are old and likely show up in the training dataset.”
> >
>
> Our benchmark implements protective measures in its current setting to ensure evaluation integrity and eliminate train-test contamination risks.
>
> Specifically, we protect evaluation files as read-only and protected from agent modification. Agents must utilize these protected evaluation files to assess their proposed methods. This safeguard prevents agents from bypassing the evaluation process by directly outputting results based on dataset knowledge potentially encoded in their underlying LLMs.

---

### Official Review · Reviewer_s9aw · 2025-11-03

**Soundness:** 2
**Presentation:** 3
**Contribution:** 2
**Rating:** 4
**Confidence:** 3

**Summary:**

The authors propose a new benchmark for evaluating AI agents on machine learning research tasks, named FML-bench. Specifically, the authors attempt to make those tasks more fundamental in terms of machine learning research. They pick eight domains (generalization, data efficiency, representation learning, continual learning, causality, robustness and reliability, privacy, and fairness and bias), each with corresponding target tasks, and define five evaluation metrics (Utility, Diversity, Academic Contribution Rate, Step Success Rate, and Cost). For the experiments, the authors employ and compare the AI Scientist, AIDE, and Claude Code. Considering these three AI agents as three prototypes with different exploration structures, based on the evaluation results with those, the authors suggest that broader explorations can be important for AI research agents.

**Strengths:**

1. Motivations
The motivation, which is to establish a machine learning research benchmark that encompasses core machine learning directions, sounds reasonable to me given existing work. While there is concurrent work (named MLGym) that aims to provide similar values but from a different angle, having this benchmark and the tasks it provides could be valuable for this domain.

2. Presentation
The submitted manuscript is easy to follow and well-presented. It contains intuitive figures and tables, which can give the readers clear ideas about what the benchmark is, how it is supposed to be different from other existing benchmarks, and what properties the AI agents that are compared in this paper have. The experimental results are also well-organized for presentation.

3. Reproducibility and potential adoption
Presenting a benchmark paper, the authors clearly reveal the prompts used and release their source codes. This can be important for not only reproducibility but also the adoption of the benchmark.

**Weaknesses:**

1. Some conclusions may need more evidence
It looks to me that some conclusions made in this work may be somewhat early conclusions without strong evidence. The authors propose that "emphasizing the breadth of exploration may lead to more effective research outcomes than focusing solely on incremental refinement," which is based on the results with the AI Scientist, AIDE, and Claude Code. Given the differences between these methods, I believe there can be other factors that have meaningfully contributed to the results, other than the breadth. Also, the authors suggest that "CLI-style agents are less suitable for automatic machine learning research than agents specifically designed for it." I think drawing this conclusion from the results with Claude Code may not be fully justified, as Claude Code is specifically designed for engineering tasks rather than research tasks.

2. Definition and implication of the evaluation metrics
I have concerns regarding how the evaluation metrics are defined and their implications. Regarding the academic contribution rate metric, I am not convinced that this is how the academic contribution of the development of some work should be evaluated. For instance, some agents might make fair academic/research contributions in only a few steps and then polish the codes or experiments by spending many small steps. Depending on the types or complexity of hypotheses or academic contributions, they may require more tuning (or "engineering") steps to achieve the desired goal. Also, the diversity metric may be susceptible to benchmark gaming/hacking.

3. Reliability of the LLM-based evaluation of academic contributions
The authors prompt the Qwen3 235b-a22b model for the evaluation of academic contribution, which can have subjectivity in it. While it is understandable that it could be challenging to define systematic or less heuristic evaluation schemes, the current state of the work lacks proper evaluation of the evaluation method. One possible way would be conducting a human study to check how the evaluation by Qwen3 235b-a22b is aligned with human experts' evaluation.

4. Minor issues
- I believe "CodeGraphBERT" should be "GraphCodeBERT."
- "Step To Target" only appears once in Appendix C.

**Questions:**

Please take a look at the weaknesses section.

---

> ### Author Response · Authors · 2025-12-02
> **Response to Reviewer  s9aw [1/2]**
>
> We thank the reviewer for the constructive feedback and address each point below:
>
> > **Q.1.1.**: “The authors propose that "emphasizing the breadth of exploration may lead to more effective research outcomes than focusing solely on incremental refinement," which is based on the results with the AI Scientist, AIDE, and Claude Code. Given the differences between these methods, I believe there can be other factors that have meaningfully contributed to the results, other than the breadth.”
> >
>
> We thank the reviewer for this important concern. We agree that multiple factors beyond exploration breadth (e.g. the academic contribution ratio we analyzed) may contribute to the observed performance differences. However, we would like to emphasize that one of the goals of our work is **providing a systematic benchmark that enables researchers to identify and analyze factors influencing agent performance.** Through our benchmark, we have identified exploration breadth as one potentially important factor for achieving effective research outcomes.
>
> To further substantiate the relationship between exploration strategy and performance, we conducted a correlation analysis between exploration diversity and outcome quality.
>
> - Our findings show that TheAIScientist, which emphasizes exploration breadth more than other agents, exhibits higher diversity scores:
>
>
>     |  | TheAIScientist | AIDE | Claude Code |
>     | --- | --- | --- | --- |
>     | diversity | 24.66 ± 8.76 | 23.50 ± 13.88 | 8.75 ± 6.07  |
> - Moreover, across five tasks, we observed consistent positive correlations between exploration diversity and performance, with p-values indicating significant, borderline significant, or suggestive evidence of this relationship. Notably, for the five tasks, higher exploration diversity was associated with better outcomes.
>
>
>     | Task | p-value | Correlation | Significance |
>     | --- | --- | --- | --- |
>     | Generalization | 0.113738 | **0.4418** | suggestive evidence |
>     | Data Eff. | 0.012058 | **0.6287** | significant |
>     | Rep. Learn. | 0.135768 | **0.4036** | suggestive evidence |
>     | Cont. Learn. | 0.000186 | **0.8374** | significant |
>     | Causality | 0.466795 | -0.2036 | not significant |
>     | Robust. & Rel. | 0.194413 | -0.3846 | not significant |
>     | Privacy | 0.647916 | -0.1286 | not significant |
>     | Fair. & Bias | 0.051652 | **0.5292** | borderline significant |
>     - p < 0.05 (significant); p < 0.10 (borderline significant); p < 0.15 (suggestive evidence); p >= 0.15 (not significant)
>     - We negated scores for tasks where lower values indicate better performance, ensuring that positive correlations consistently represent improved performance with increased diversity.
>
> We will update the manuscript accordingly to better clarify and elaborate on these findings
>
> > **Q.1.2:** “Also, the authors suggest that "CLI-style agents are less suitable for automatic machine learning research than agents specifically designed for it." I think drawing this conclusion from the results with Claude Code may not be fully justified, as Claude Code is specifically designed for engineering tasks rather than research tasks.”
> >
>
> We thank the reviewer for raising this concern. Our goal is to compare the performance of general-purpose agents with research-specific agents in the research setting. Because **general-purpose agents offer potential advantages over agents specifically designed for research tasks**: they can autonomously leverage more diverse and flexible tools (such as web search queries and autonomous script generation for analysis), and they benefit from superior programming and reasoning capabilities due to continuous industrial development and refinement. Therefore, we sought to compare their performance on research problems.

---

> ### Author Response · Authors · 2025-12-02
> **Response to Reviewer  s9aw [2/2]**
>
> > **Q.2.1. Concern about academic contribution rate metric:** “I have concerns regarding how the evaluation metrics are defined and their implications. Regarding the academic contribution rate metric, I am not convinced that this is how the academic contribution of the development of some work should be evaluated. For instance, some agents might make fair academic/research contributions in only a few steps and then polish the codes or experiments by spending many small steps. Depending on the types or complexity of hypotheses or academic contributions, they may require more tuning (or "engineering") steps to achieve the desired goal.”
> >
>
> We thank the reviewer for identifying this lack of clarity in our description.
>
> To clarify, the **academic contribution rate metric compares each step with the baseline method rather than comparing consecutive steps with each other**. This metric is designed to encourage agents to propose new ML methods that differ from the baseline rather than merely tuning hyperparameters (engineering modification). Any method that introduces a new machine learning idea (e.g. a novel objective or inductive bias) is classified as an academic modification, regardless of its complexity.
>
> Regarding the reviewer's concern, since we compare each step against the baseline, **both initial new ML method proposals and subsequent refinement steps would be classified as academic contributions**. Specifically, if step_k proposes a new method and is classified as an academic step, then step_k+1, which refines step_k, is also classified as an academic step because both are compared against the baseline rather than against each other.
>
> We will revise the manuscript to provide a clearer description of this metric and its interpretation.
>
> > **Q.2.2. Concern about diversity metric**: “Also, the diversity metric may be susceptible to benchmark gaming/hacking”
> >
>
> We thank the reviewer for raising this concern. To address this, we will clarify in the manuscript that the **diversity metric serves as an analytical tool** for characterizing agent exploration behavior rather than an objective, thereby mitigating concerns about benchmark gaming or hacking.
>
> Moreover, as discussed in our response to Q.1.1., the diversity metric provides valuable insights into agent characteristics. Our analysis reveals that exploration diversity positively correlates with performance across 5 tasks, with higher diversity associated with improved outcomes.
>
> > **Q.3: Human evaluation of LLM’s judgement:** “The authors prompt the Qwen3 235b-a22b model for the evaluation of academic contribution, which can have subjectivity in it. While it is understandable that it could be challenging to define systematic or less heuristic evaluation schemes, the current state of the work lacks proper evaluation of the evaluation method. One possible way would be conducting a human study to check how the evaluation by Qwen3 235b-a22b is aligned with human experts' evaluation.”
> >
>
> We thank the reviewer for this valuable suggestion. As discussed in our response to Q.1.2., the academic contribution rate metric compares each step with the baseline method rather than comparing consecutive steps with each other, and is designed to encourage agents to propose novel machine learning methods that differ from the baseline rather than merely tuning hyperparameters (engineering modifications).
>
> Following the reviewer’s recommendation, we conducted a human evaluation to validate the accuracy of Qwen3-235b-a22b's judgments. Three human annotators each evaluated 200 samples (5 agents × 40 samples per agent). The results are presented in the following table:
>
> | Model | Accuracy of LLM Evaluation (Human-evaluated) | LLM-Judged Academic Contribution Ratio | Human-Judged Academic Contribution Ratio |
> | --- | --- | --- | --- |
> | TheAIScientist - GPT-5 | 0.7982 ± 0.0992 | 0.83 ±  0.27 | 0.8947 ± 0.1489 |
> | TheAIScientist - Gemini-2.5-pro | 0.7807 ± 0.0969 | 0.78 ±  0.24 | 0.7281 ± 0.1183 |
> | AIDE - GPT-5 | 0.8198 ± 0.1215 | 0.84 ±  0.20 | 0.7928 ± 0.1770 |
> | AIDE - Gemini-2.5-pro | 0.8120 ± 0.0604 | 0.65 ±  0.34 | 0.6239 ± 0.0944 |
> | ClaudeCode - Opus-4.1 | 0.7701 ± 0.1445 | 0.25  ± 0.36 | 0.5057 ± 0.1445 |
> | Overall mean | 0.7962 ± 0.0186 | 0.67 ± 0.2206 | 0.7091 ± 0.1345 |
> - Our human evaluation reveals that Qwen3 235b-a22b achieves an overall accuracy of 79.62% in its judgments.
> - Additionally, we report the human-evaluated academic contribution ratios in the table above, providing a more reliable assessment of agent performance.
>
> We will incorporate these human evaluation results into the manuscript accordingly.
>
> > **Q.4.** “I believe "CodeGraphBERT" should be "GraphCodeBERT."; "Step To Target" only appears once in Appendix C.”
> >
>
> Thank you for pointing this out. We will correct the model name and update Appendix C accordingly.

---

### Author Response · Authors · 2025-12-03
**General Response to All Reviewers**

We sincerely thank all reviewers for their thorough evaluation and constructive feedback. We are encouraged by the recognition of our benchmark's motivation to assess scientific research competence rather than engineering ability (s9aw, NpRn), the comprehensive evaluation metrics with exploration diversity and academic contribution rate providing analytical tools to understand what agents are doing (mYE9, 9hah), and our efforts in ensuring reproducibility and community accessibility (s9aw, NpRn, mYE9).

\
Our benchmark aims to address a critical gap in the ML research agent evaluation landscape. Existing benchmarks in this domain remain insufficiently explored, with:

- prior work predominantly emphasizing agents' engineering capabilities through Kaggle-style competitions and use-case-oriented tasks. (line#50-53)
- existing research agent benchmarks (including MLE-bench, MLAgentBench, DSBench, and ML-Dev-Bench) primarily focus their evaluation on final task performance metrics (e.g., accuracy, recall) and computational cost, while overlooking the characteristics of agents' internal iterative processes. (line#53, #75)

FML-bench represents an important step forward in two key aspects:

- we construct a benchmark comprising 8 diverse and fundamental ML problems designed to assess agents' capabilities in tackling fundamental machine learning research problems. (line#84-87)
- we provide a systematic evaluation platform enabling researchers to identify and analyze factors influencing agent performance. Specifically, our benchmark introduces exploration Diversity and Academic Contribution Rate metrics to analyze agent behavior throughout the entire research iteration process, providing insights into how agents explore and develop solutions rather than solely evaluating their final outcomes. (line#103-107, #125-129, Sec. 5.3.2 & 5.3.3)

Through empirical studies on our benchmark, we have identified exploration breadth and diversity as factors influencing final performance:

- Agents emphasizing exploration breadth demonstrate more effective research outcomes (e.g., TheAIScientist achieves the highest number of best performances across tasks). (Sec. 5.3.1)
- Agents prioritizing exploration breadth exhibit higher diversity. (Sec. 5.3.2)
- Higher diversity correlates with better performance in 5 out of 8 tasks. (Sec. 5.3.2; Response to Reviewer NpRn [2/4] - Q.4.)

Our work advocates for examining agents' internal behaviors and iterative processes, moving beyond simple comparisons of final task performance. We envision that FML-Bench will help the community identify which agent characteristics influence research performance, ultimately advancing the development of research agents.

\
We have addressed the reviewers' concerns in our detailed responses. Here we summarize the main actions taken:

1. **Human evaluation for academic contribution rate**: we have supplemented our evaluation with human assessment to improve the accuracy of academic contribution rate measurement. We also report the human-evaluated academic rate versus LLM-judged rate.
2. **Enhanced task difficulty**: We have increased the difficulty of 3 tasks (Fairness & Bias, Causality, Robustness & Reliability) to better differentiate agent performance and provide headroom for future improvements.
3. **Correlation analysis**: We have recalculated the correlation between diversity and outcomes on the updated benchmark with a larger sample size, enhancing the robustness of our analysis.
4. **Best case and worst case analysis**: We have added best case and worst case analysis of agent results to provide deeper insights into agent and LLM performance characteristics on FML-Bench.
5. **Clarifications on evaluation concerns**:
    - **Academic Contribution Rate**: This metric is designed to encourage agents to propose new ML methods that differ from the baseline rather than merely tuning hyperparameters (engineering modification). It compares each step with the baseline method rather than consecutive steps. Therefore, both initial methodological proposals and subsequent refinement steps are classified as academic contributions.
    - **Diversity metric**: We employ GraphCodeBERT, which analyzes code structure and data-flow graphs rather than surface tokens. Code snippets with identical logic but different variable names receive high similarity scores and will be treated as low diversity.
    - **Evaluation objective**: Following the reviewers' suggestions, we will revise the evaluation framework as follows: Utility serves as the sole metric reflecting the final performance objective, while the other metrics (including Diversity and Academic Contribution Rate) serve as descriptive and analytical indicators.
    - **Evaluation integrity**: Our benchmark protects evaluation files as read-only and prevents agent modification to ensure evaluation integrity. Agents must utilize these protected evaluation files to assess their proposed methods.

---

### Meta-Review · Area_Chair_tWu3 · 2026-01-02

**Summary:**

This submission proposes FML-bench, a benchmark for evaluating automatic ML research agents on eight “fundamental ML” tasks instantiated from real-world repositories, and reports results/analyses for several representative agent frameworks with a five-metric evaluation suite (utility, diversity, academic contribution rate, cost, step success).

Reviewers acknowledged  the motivation (shifting beyond purely “engineering execution”), the practical repo-based setup, and generally clear presentation and potential usefulness of this paper. However, they raised major concerns that materially limit the benchmark’s claims: conclusions about “exploration breadth” may be driven by the specific design and could incentivize shallow combinations; several metrics (notably “academic contribution rate”) are not well-grounded and include subjective elements; and coverage/generalizability is weak because broad categories are represented by single tasks, with additional worries about rigor and longevity.

The rebuttal clarifies some evaluation details (e.g., not aggregating metrics; reframing ACR), but does not fully address the incentive/validity issues or the limited scope. While the work is timely, the current benchmark does not convincingly differentiate itself versus numerous existing benchmarks in scale, problem formulation, or evaluation strength. I recommend rejecting this paper.

**Reviewer Concerns:**

The outstanding concerns after rebuttal.
- The benchmark’s incentive structure behind the breadth claim,
- The limited scope/coverage (still 8 tasks, single-repo per task),
- The broader concerns about benchmark longevity and generality.

**Reviewer Scores:**

Scores are overall borderline (4/4/4/6).

---

### Decision · Program_Chairs · 2026-01-26

Reject